# Integrating single-cell and spatially resolved transcriptomic strategies to survey the astrocyte response to stroke in male mice

Erica Y. Scott [1,2,3], Nickie Safarian[4,5], Daniela Lozano Casasbuenas[1,6], Michael Dryden [2,3], Teodora Tockovska[6], Shawar Ali[6], Jiaxi Peng[2], Emerson Daniele[1,7], Isabel Nie Xin Lim[1], K. W. Annie Bang [8], Shreejoy Tripathy[4,5,10], Scott A. Yuzwa[6,10], Aaron R. Wheeler [2,3,9,10] & Maryam Faiz [1,6,7,10] ✉

Astrocytes, a type of glial cell in the central nervous system (CNS), adopt diverse states in response to injury that are influenced by their location relative to the insult. Here, we describe a platform for spatially resolved, single-cell transcriptomics and proteomics, called tDISCO (*tissue*-digital microfluidic isolation of single cells for -Omics). We use tDISCO alongside two high-throughput platforms for spatial (Visium) and single-cell transcriptomics (10X Chromium) to examine the heterogeneity of the astrocyte response to a cortical ischemic stroke in male mice. We show that integration of Visium and 10X Chromium datasets infers two astrocyte populations, proximal or distal to the injury site, while tDISCO determines the spatial boundaries and molecular profiles that define these populations. We find that proximal astrocytes show differences in lipid shuttling, with enriched expression of *Apoe* and *Fabp5*. Our datasets provide a resource for understanding the roles of astrocytes in stroke and showcase the utility of tDISCO for hypothesis-driven, spatially resolved single-cell experiments.

The high-throughput study of single-cell transcriptomes has become routine for the identification of cell state and type in healthy tissues and during injury/disease[1]. Unfortunately, most single-cell techniques evaluate cells in suspension, which ignores the vast phenotypic data that can be gathered from a cells' spatial position within a tissue. This spatial information is particularly important in the study of positional injuries or pathologies, as the physical proximity of a cell to the injury influences the cell's transcriptome.

Astrocytes are a glial cell type in the central nervous system (CNS). In response to CNS disease and injury, astrocytes adopt diverse states that are characterized by unique molecular profiles and functions[2]. To add further complexity, the heterogeneity of the astrocyte response is influenced by many variables, including injury/disease stage, severity, and/or distance from a pathological event[3]. Therefore, the analysis of astrocytes within their context is key to understanding their diverse response to injury.

[1]Department of Surgery, University of Toronto, 1 King's College Circle, Toronto, Ontario M5S 1A8, Canada. [2]Department of Chemistry, University of Toronto, 80 St. George St., Toronto, ON M5S 3H6, Canada. [3]Donnelly Centre for Cellular and Biomolecular Research, University of Toronto, 160 College Street, Toronto, ON M5S 3E1, Canada. [4]Department of Psychiatry, University of Toronto, 250 College St., Toronto, Ontario M5T 1R8, Canada. [5]The Krembil Centre for Neuroinformatics, Centre for Addiction and Mental Health, Toronto, ON, CA, 250 College St., Toronto, Ontario M5T 1R8, Canada. [6]Department of Laboratory Medicine & Pathobiology, University of Toronto, 1 King's College Circle, Toronto, Ontario M5S 1A8, Canada. [7]Institute of Medical Sciences, University of Toronto, 1 King's College Circle, Toronto, Ontario M5S 1A8, Canada. [8]Lunenfeld-Tanenbaum Research Institute, Flow Cytometry Core, Sinai Health, Toronto, Ontario M5G 1X5, Canada. [9]Institute of Biomedical Engineering, University of Toronto, 164 College St., Toronto, ON M5S 3G9, Canada. [10]These authors jointly supervised this work: Shreejoy Tripathy, Scott A. Yuzwa, Aaron R. Wheeler, Maryam Faiz. ✉e-mail: maryam.faiz@utoronto.ca

Here, we examined the spatial and single-cell heterogeneity of astrocytes following a cortical ischemic stroke. Three tools were utilized to probe this brain injury at multiscale resolution, as shown in Fig. 1. First, Visium was used to establish a spatially resolved overview (but lacking single-cell resolution) of the molecular signatures associated with the ischemic cortex. For this, we chose a time course that represents different pathophysiological processes characteristic of the acute (d2), subacute (d10), and chronic (d21) stages of stroke[4–6]. Second, 10X Chromium was applied to suspensions of cells from injured tissues to study the heterogeneity of astrocytes in subacute stroke. Finally, the recently reported digital microfluidic isolation of single cells for -Omics[7] (DISCO) technique, distinct from the tissue clearing DISCO technology[8], was applied to evaluate individual astrocytes' transcriptomes and proteomes at various distances from the infarct site in the subacute stage. This required a complete re-design of the DISCO system (originally demonstrated only with cultured cells), focusing on facile selection of cells from tissue sections, combined with improved automation and imaging capabilities. We call the technique tissue-DISCO (tDISCO) and propose that it will be useful for a wide range of applications going forward. Altogether, the data reported here show the acquisition of different astrocytic states after stroke that are influenced by time and proximity to the stroke lesion site while unveiling the utility of a high-content single-cell tool called tDISCO.

## Results

### Spatial transcriptomics reveals the molecular signatures of stroke in space and time

To understand the molecular profiles of stroke in space and time, we performed spatial transcriptomics with 10X Visium. A focal cortical stroke model that results in small, localized, reproducible ischemic lesions[9] was used to examine three biological replicates of the stroke-injured hemisphere at acute (d2), sub-acute (d10), and chronic (d21) time points, and following a control sham injury (d2 sham) (Supplementary Fig. 1).

To identify the gene signatures associated with each stage of stroke we used non-negative matrix factorization (NMF). This approach uncovered factors (features) that were specifically associated with the lesion site at d2 and d10 (Fig. 2A). The genes driving these factors broadly showed a macrophage dominant signature (i.e., Factor 1) in the acute phase (d2), while a generic glial signature (i.e., Factor 2) was predominant in the subacute phase (d10) (Fig. 2B). At

d21, no factor was seen to localize around the injury site, rather there was a factor indicative of re-establishment of normal cortical architecture (i.e. Factor 3, Fig. 2A, bottom row). To further resolve genes driving these factors, we mapped the gene signatures associated with d2, d10, and d21 (Fig. 2B) back onto the Visium brain sections and found that they recapitulated the spatial localization of the NMF factors (Fig. 2C).

Next, we used UMAP dimension reduction to observe discrete, spatially localized gene expression profiles from all the post-stroke tissues (d2, d10, and d21). Again, we were able to identify two distinct clusters associated with the injured cortex at d2 and d10 (Fig. 2D, E). Of interest, the d2 gene expression signature was restricted to d2 (Fig. 2E, second row). In contrast, the d10 signature was found within the lesion site at d10 but was also found distal to the injury site at d2 and d21 (Fig. 2E, bottom row). To better delineate the gene expression associated with the injury site, we analyzed how the spots along the "border" of the injury site differed transcriptionally. At d2, differential gene expression at the lesion edge versus the injury site showed a consistent expression of markers associated with macrophages, such as *Lyz2*[10], and *Lgals3*[11] in the injury site (dark purple). In contrast, the d2 border spots (grey) expressed astrocytic and microglia markers such as *Gfap*[12], *C4b*[12], and *Lcn2*[13] (Fig. 2F). At d10, the injury site (yellow) was characterized by microglia and astrocyte-related genes such as *Cst7*, *Tyrobp*, *C1qc*[14], and *Hexb*[15], while the d10 border spots (navy blue) showed expression of *Aldoc*, and *Clu* (Fig. 2G). To understand the cell types associated with these molecular signatures, we used cell type markers derived from the Allen Brain Atlas to deconvolve the cell types within the Visium data (Supplementary Fig. 2). We found that macrophage markers characterized the d2 infarct-related expression signature, while glial cell types dominated the d10 profile. At d21, layer-specific projection neuron signatures were most prominent (Supplementary Fig. 2). Similar clustering and gene markers were also found in our two other biological replicates at each time point (Supplementary Fig. 3).

Our analyses show that the molecular profiles of the stroke-injured cortex evolve over time and in space. Our data suggests that in the acute stage (d2), infarct site expression is largely dominated by macrophage markers. In the subacute stage (d10), reactive glial signatures are prominent in the injury site. At the chronic time point (d21), we found restoration of the normal cortical architecture, as defined by gene expression.

### Integrating spatial and single-cell transcriptomics identifies the heterogeneity of the astrocyte response in subacute stroke

As we found a glial dominant signature in our subacute (d10) Visium datasets, we chose to study the astrocyte response at this stage. In subacute stroke, astrocytes play important roles in angiogenesis[16] and synaptogenesis[17], but the molecular profiles and degree of heterogeneity are largely unknown. Therefore, we used 10X Chromium to perform single-cell RNA-seq on astrocytes isolated from stroke-injured and uninjured cortices in two independent experiments.

The first set of cells was isolated from stroke-injured and uninjured control tissue using MACS against ACSA-2. ACSA-2 is an astrocyte cell surface antigen expressed on the surface of GLAST+ astrocytes in the sensory-motor cortex (Supplementary Fig. 4A). MACS was followed by FACS selection of live nucleated (DRAQ5 + ) ACSA2-PE+ astrocytes. Within the population of DRAQ5 + DAPI- cells, 93% of cells were ACSA2-PE + , with either intermediate (ACSA2int, 46.08%) or high (ACSA2hi, 46.91%) fluorescence intensity (Supplementary Fig. 4A). We retrieved a total of 13,517 ACSA-2+ cells, including 6243 injured and 7274 uninjured cells (Fig. 3A). To our surprise, the gene expression of these cells was dominated by microglial rather than astrocytic markers (Fig. 3B). We then attempted to identify a subpopulation within this set with more astrocyte-like expression and delineated 2856 cells that expressed

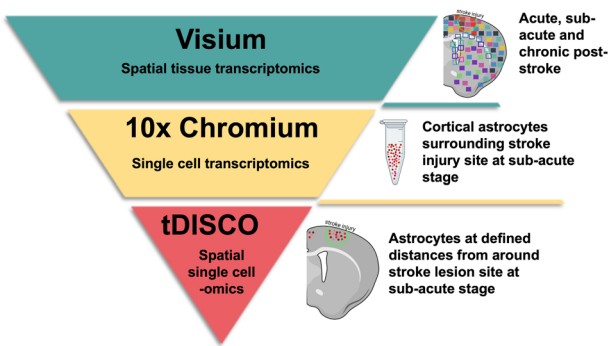

**Fig. 1 | Schematic of sequencing technologies used to study astrocytes in stroke.** A broad overview of stroke over time (d2 acute, d10 subacute, d21 chronic) was performed using spatially resolved tiles (consisting of multiple cells) with the commercial 10x Visium. Next, astrocyte heterogeneity was examined in suspensions of individual astrocytes isolated from the stroke-injured cortex in the sub-acute stage using 10x Chromium. Finally, to spatially resolve the molecular profiles of individual astrocytes in sub-acute stroke, single astrocytes were captured at various distances from the injury site and analyzed using tDISCO. Biorender was used to create the schematic.

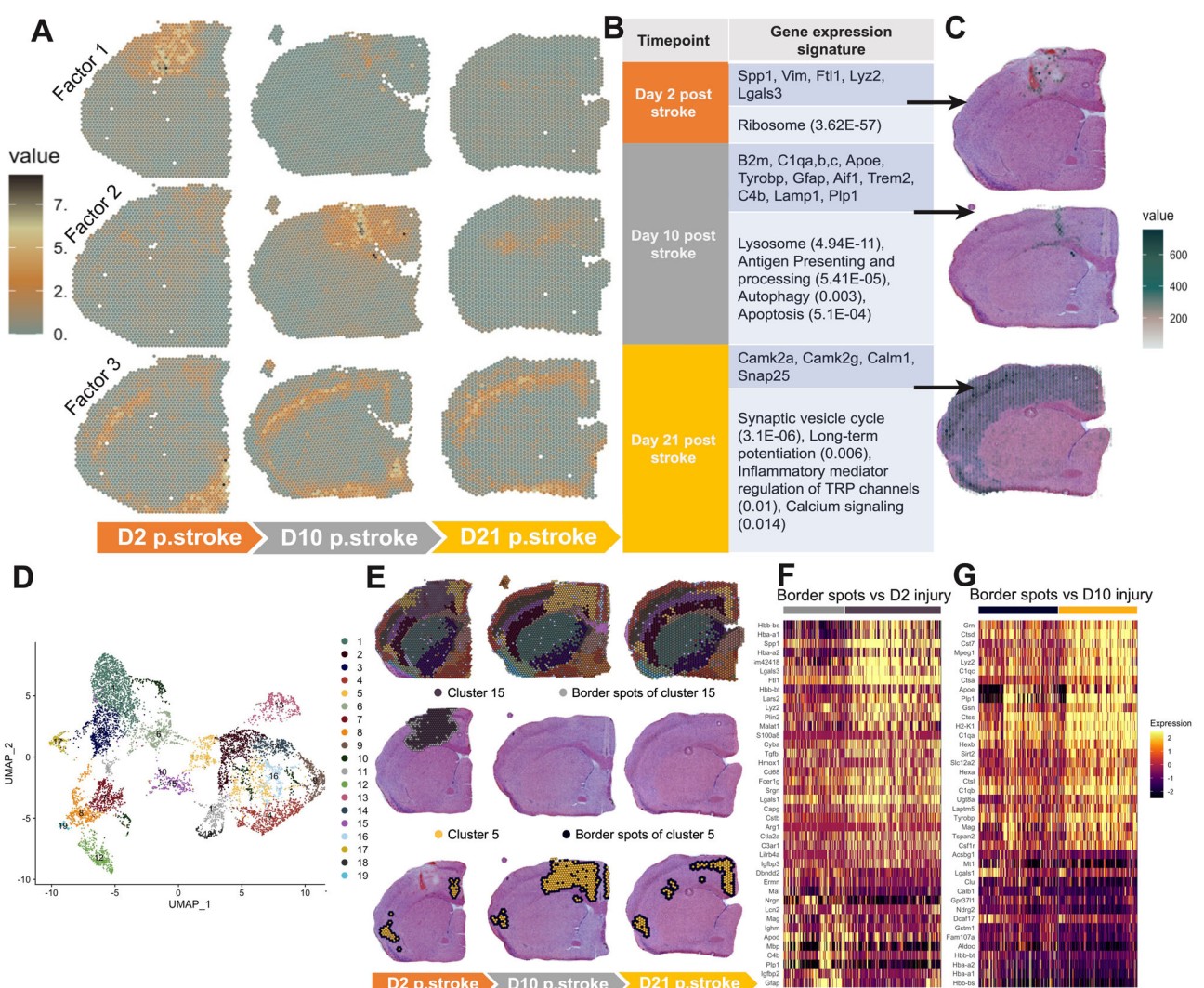

**Fig. 2 | Shifts in broad spatial transcriptional profiles across time post-stroke.**
**A** Visium d2, d10, and d21 brain sections (columns) with overlaid NMF-identified factors (rows) that specifically localized to the injury area at d2 (factor 1, top row) or d10 (factor 2, middle row). No factor localized to the injury area at d21, instead a factor that follows the pattern of cortical neuronal layering was observed (factor 3, bottom row). Values plotted are spatial autocorrelation values of the factors per spot. **B** Table of select genes (dark blue) driving each of the factors in (**A**) and the pathways associated with the genes (light blue). Pathways were obtained by running the genes through EnrichR and adjusted p-values are shown in parentheses. **C** Visium d2 (top), d10 (middle) and d21 (bottom) sections with overlaid genes from (B) that drive factors 1, 2, and 3 from (**A**). The value represents the summed SCT normalized expression values corresponding genes from (**B**). **D** UMAP plot of all the spots from Visium d2, d10, and d21 sections in a UMAP-reduced dimension. Colors for clusters remain constant through (**E–G**). **E** Visium d2, d10 and d21 sections overlaid with all clusters from (**D**) are shown in the top row. Overlay of individual clusters associated with the d2 injury space and d10 injury space are shown in the middle and bottom rows, respectively. Spots bordering these clusters (grey spots in the middle row, navy blue in the bottom row) were isolated for differential gene expression. **F, G** Heatmaps representing the differential gene expression between the injured clusters in d2 (**F**) and d10 (**G**) and their neighboring border spots (colors correspond to spot colors in **E**). Expression represents SCT normalized expression values where high is bright yellow and low is dark purple. Source data are provided as a Source Data file.

*Aif1* and *Trem2* at levels below a normalized expression value of 0.1 (Supplementary Fig. 5). To understand the location of these astrocytic cells within the stroke injured cortex, we overlaid them onto the d10 post-stroke Visium section (Fig. 3C). The cells showed two distinct localization patterns either within the stroke infarct site (Cluster 6) or around the injury site (Clusters 3, 5, 7, and 9) (Fig. 3C).

The finding in Fig. 3C was intriguing. Therefore, to further examine the spatial localization of these cells, we then examined a set of purposefully selected astrocytes (rather than a stratified sub-population). DRAQ5 + DAPI-ACSA2+ cells were purified from cortical tissue using our MACS/FACS strategy. However, this time, we used a more stringent criterion for astrocyte selection and only collected ACSA2hi astrocytes (43.11% of the total DRAQ5 + DAPI-ACSA2-PE+ cells) for downstream 10X Chromium. A total of 5695 cells were retrieved in this ACSA2hi set,

including 2492 injured and 2993 uninjured cells. While these transcriptomes did not form discrete clusters (Fig. 3D), they did show a much more astrocyte-like pattern of expression, except for cluster 6, which showed expression for vascular and leptomeningeal cells (VLMCs)[18] (Fig. 3E). We then used proportion analysis to identify cell clusters that were biased toward uninjured or stroke samples and found that cluster 5 was predominant in stroke-injured samples (Fig. 3F). However, when we overlaid all clusters onto the d10 Visium section, cluster 5 did not overlap with the d10 section (Fig. 3G). This highlighted the flaws associated with inference used when integrating 10X Chromium datasets with d10 Visium sections. Interestingly, we were able to find spatial patterns of the remaining cells (that did not form meaningful clusters in Fig. 3D). These clusters were found within and around the stroke injury site at d10. We termed the astrocytes localized within the

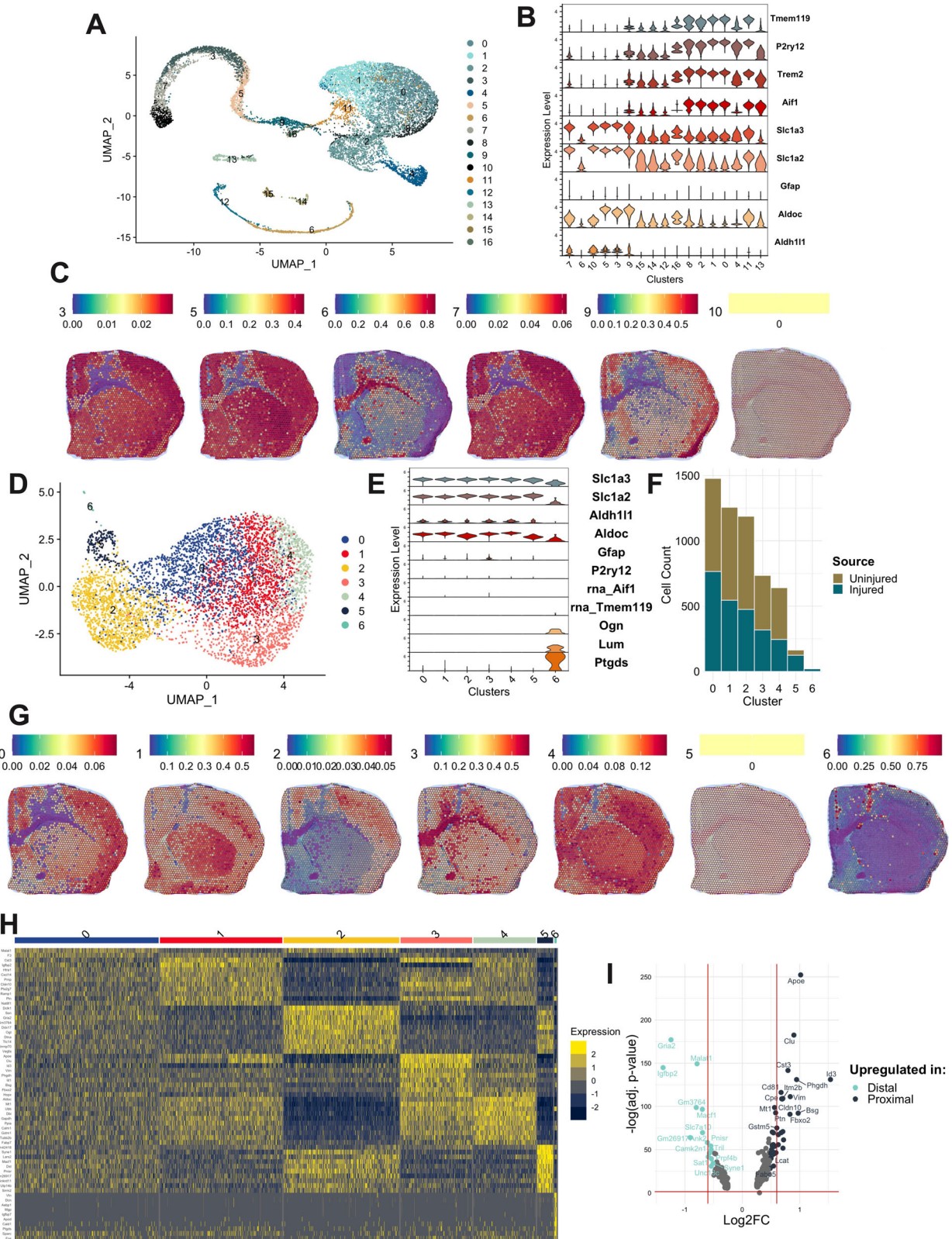

infarct, proximal, and astrocytes devoid of this area, distal [to the infarct] (Fig. 3G). The top 10 genes marking each cluster in Fig. 3D are highlighted in Fig. 3H. Genes such as *Syne1*, *Lars2*, and *Macf1* define Cluster 5 astrocytes.

Next, we specifically examined the differentially expressed genes between cluster 3 and cluster 0, which this analysis suggested as putative proximal and distal astrocytes, respectively, and found 392

genes (adjusted *p*-value < 0.05). Among these genes were *Apoe*, *Clu*, *Cd81*, *Id3*, and *Fabp5*, which were significantly upregulated in proximal cells (Fig. 3I). The genes *Fabp5* (adjusted *p*-value = 7.63 × 10⁻¹⁷¹), *Apoe* (adjusted p-value = 3.87 × 10⁻⁸²), *Cd81* (adjusted p-value = 3.91 ×10⁻¹⁸¹), and *Clu* (adjusted *p*-value = 2.16 × 10⁻²⁵⁷) were also enriched in the putatively proximal cluster 6 identified in the smaller astrocytic data-set (Fig. 3C, Supplementary Data 1), contributing to the robust nature

**Fig. 3 | Combining the granularity of single cell transcriptomics with spatial information from Visium uncovers the heterogeneity of astrocytes in sub-acute stroke. A** UMAP plot of 13,517 ACSA-2+ cells from injured (6243 cells) and non-injured (7274 cells) cortices. **B** Violin plots showing the distribution of microglial (*Tmem119, P2ry12, Trem2, & Aif1*) and astrocytic (*Slc1a3, Slc1a2, Gfap, Aldoc, Aldh1l1*) genes (rows) across similarity-sorted clusters (columns, sorted by average gene expression) that are derived from the UMAP in (**A**). Expression (y-axis) values represent SCT-normalized gene expressions. **C** Visium d10 section with overlaid astrocyte clusters (cells with expression levels of Aif1 and Trem2 less than 0.1). Values in the colourmap represent prediction scores of the clusters on the Visium spots. **D** UMAP plot of a 5695 ACSA-2+ hi cells from injured (2492 cells) and non-injured (2993 cells) cortices. **E** Violin plots showing the distribution of microglial (*Tmem119, P2ry12, Trem2, & Aif1*), astrocytic (*Slc1a3, Slc1a2, Gfap, Aldoc, Aldh1l1*), and VLMC (*Ogn, Lum, & Ptgds*) genes (rows) across clusters (columns) that are derived from the UMAP in (**D**). Expression (y-axis) values represent SCT-normalized gene expressions. Gene markers that are absent compared to (**B**) is due to the gene not being captured in this dataset. The gene prefix "rna_" indicates that the gene was only found in raw (not normalized) gene counts. **F** Histogram depicting the number of cells belonging to uninjured versus stroke-injured samples across each cluster from (**D**). **G** Astrocytic clusters from (**D**) overlaid onto the D10 Visium section. Values in colourmap represent prediction scores of the clusters on the Visium spots. **H** Heatmap of the top 10 genes (rows) enriched in each cluster (columns) highlights substantial gene expression overlap, likely contributing to small distances between UMAP-derived clusters in (**D**). Expression value represents log-normalized and scaled expression values. **I** Volcano plot between cluster 3 (inferred proximal) versus 0 (inferred distal). The higher the gene (dot) the lower the p-value. Genes upregulated in proximal astrocytes are shown in dark blue (positive). Genes upregulated in distal astrocytes are shown in light blue (negative). Source data are provided as a Source Data file.

of these candidate genes in astrocytes present within the stroke infarct area.

Our analysis showed that understanding the heterogeneity of astrocyte response to stroke required integration and inference of both 10X Chromium and Visium datasets. The 10X Chromium platform provided large numbers of single-cell astrocyte transcriptomes but without a spatial context. When combined with Visium, we were able to infer the relative spatial properties of these astrocytes. This allowed us to identify transcriptionally different populations of astrocytes located proximal and distal to the stroke infarct site and may reflect functional differences related to their position relative to the stroke infarct. For example, genes associated with proximal astrocytes encompass fatty acid and cholesterol metabolism (*Apoe, Fabp5,* and *Clu*), important for astrocyte-neuron metabolic coupling in response to ischemia[19]. These integration strategies were useful for identifying spatially distinct astrocyte populations, however, the lack of placement for cluster 5 highlighted the need for technology that can directly measure a single cell transcriptome as it relates to a specific spatial location.

**Spatially resolved single-cell transcriptomics with tDISCO defines proximal and distal astrocytes**

To further study proximal and distal astrocytes with single-cell resolution and direct spatial measurements, we developed a spatially resolved single-cell -omics platform called tDISCO (Fig. 4A, B). tDISCO represents a substantial improvement on its predecessor[7] in two critical categories – (i) a new interface and method was developed to permit the selection of single cells from cryosectioned tissue slices (Supplementary Fig. 6), and (ii) the instrument was redesigned to allow for end-to-end automation of the tDISCO process. The tDISCO system makes use of a laser to create cavitation bubbles that mechanically release selected single cells into the receiving buffer above the tissue, while digital microfluidics is used to collect this single cell and replenish the buffer droplet above the tissue, allowing for iterative lysis of individual cells from a single tissue. The tDISCO system allowed us to directly distinguish between proximal and distal astrocytes using transcriptomics and proteomics.

To ensure specificity and minimal crosstalk between cells collected by tDISCO, we isolated NEUN+ neurons and GFAP+ astrocytes within close proximity of one another (Fig. 4C). Downstream scRNA-seq (Fig. 4D) or scProteomics (Fig. 4E) on the captured cell lysates showed appropriate astrocytic gene expression in GFAP+ astrocytes, with less expression in NEUN+ cells. Neuronal gene expression was not as discrete between NEUN+ and GFAP+ cells (Supplementary Fig. 7), perhaps due to the collection of neuronal pieces when lysing or because these neuronal transcripts are expressed in astrocytes.

We then used tDISCO to survey astrocytes as a function of distance from the injury site. We captured astrocytes from the injured cortex in subacute stroke (d10) for downstream scRNA-seq. We used GFAP to identify astrocytes within the stroke injured cortex, as GFAP is well known to be upregulated after stroke[20], and was also seen in our d10 Visium experiments to define the injured cortex (Fig. 2). We lysed GFAP+ astrocytes from four different zones at increasing distances (200 μm) from the lesion site: zone A (0-200 μm), zone B (200-400 μm), zone C (400-600 μm), which we initially considered as proximal astrocytes, and in a region far from the lesion site (zone D), which we considered distal astrocytes (Fig. 4F). We were surprised by transcriptional differences between zone B and zone C cells, as we initially considered both proximal. Genes enriched in zone B astrocytes (Fig. 4G, group "a", likelihood ratio test, FDR < 0.05) encompassed pathways such as neurotrophin signaling (KEGG 2021-Human, adjusted p-value = 0.001) and MAPK signaling (KEGG 2021-Human, adjusted p-value = 0.001). Genes absent in zone B astrocytes (Fig. 4G, group "b", likelihood ratio test, FDR < 0.05) were not significantly associated with any pathways. Some of these genes, such as *Vcp*[21], are neuroprotective, and some are involved with small RNA involved in cell stress, such as *Ice1*[22]. When we examined astrocytes in zone C and D, we found genes such as *Chst3, Gan,* and *Tprkb* to be upregulated (Fig. 4G, group "c", likelihood ratio test, FDR < 0.05). Lastly, we were able to detect lncRNA (for example, Gm44432, Gm44039) (Fig. 4G, group "d"). However, these were largely not expressed in zone B cells (Fig. 4G, group "d"), suggesting that zone B astrocytes may utilize lncRNA differently than distal astrocytes. We also found differences between zone A and distal zone C/D cells (Supplementary Fig. 8).

To further understand differences in proximal (zone B) and distal astrocytes (zone D), we used tDISCO to perform single-cell untargeted mass spectrometry. There were 317 proteins detected across all samples, with fewer proteins detected in zone B cells compared to zone D cells. Proteins more abundant in proximal astrocytes (Fig. 4H, zone B, top group, p-value < 0.05) included RNH1, known to invoke translation under cell stress[23]. Other proteins in proximal astrocytes, included DSP, JUP, and DSG1A, which are desmosome proteins that may be involved in cell-cell contact and adhesion[24] (Fig. 4H, top group, p-value < 0.05). This could reflect gap junctional remodeling and communication between astrocytes in response to a CNS insult[25,26]. Among the 38 proteins upregulated in the distal astrocytes (Fig. 4H, bottom group, p-value < 0.05), pathways such as canonical glycolysis (GO:0061621, adj. p-value < 0.01), hexose biosynthetic process (GO:0019319, adj. p-value < 0.01) and glucose metabolic process (GO:0006006, adj. p-value < 0.01) were enriched. Additionally, these distal astrocytes are enriched for CFL1 and STMN1, which destabilize structural actin and microtubules, respectively[27,28]. The alpha- and beta- synucleins (SNCA & SNCB) were also both enriched in the zone D distal astrocytes.

In summary, the unique attributes of tDISCO allowed the identification of two molecularly distinct astrocyte populations that adopt a gene expression and proteomic signature based on their proximity to the lesion site: proximal zone A and zone B astrocytes and distal zone C and D astrocytes. We hypothesize that zone B proximal astrocytes play

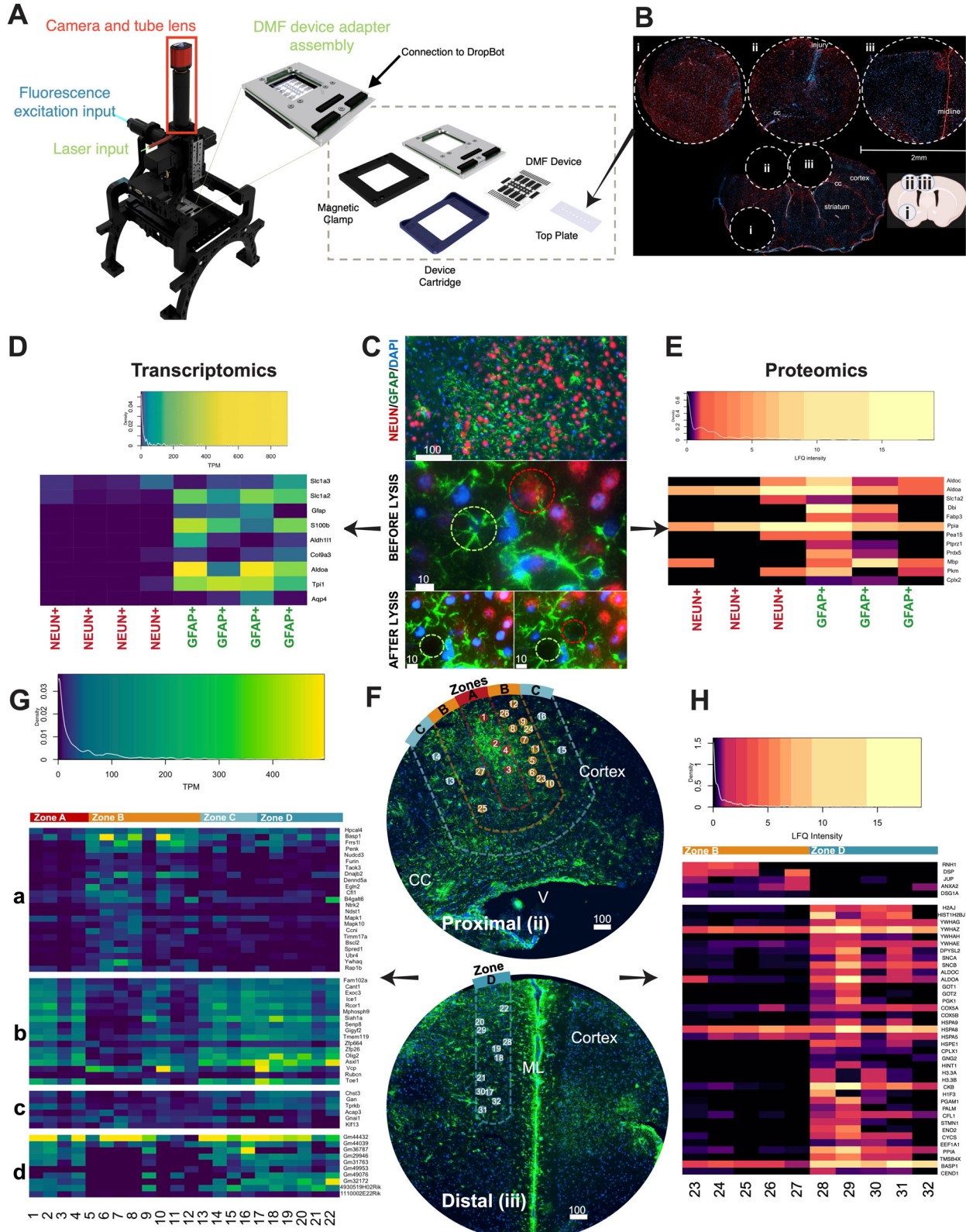

roles in circuit plasticity and synaptogenesis, while zone C and D distal cells contribute to plasticity via changes in intermediate filament networks and the extracellular matrix (which also aligns with the proteomics data), as seen in other studies involving these same genes[29,30]. Proteomics also suggests that distal and proximal astrocytes differ in their energy mobilization strategies.

## Candidate gene inconsistencies across three sequencing platforms

The 10X Visium, 10X Chromium, and tDISCO analyses generated a list of interesting candidate transcripts that appeared to be differentially expressed in astrocytes proximal and distal to stroke injury, including the candidate genes, *Apoe*, *Fabp5*, *Clu*, *Aldoc*, and *Cd81*,

**Fig. 4 | tDISCO resolves the boundaries and molecular profiles of proximal and distal astrocytes in sub-acute stroke. A** Schematic of the tDISCO system. tDISCO is comprised of an upright microscope equipped with a digital microfluidic (DMF) capable stage (grey, dashed inset), a fiber optic laser input and LED fluorescence capabilities. **B** Image shows three 2 mm tissue biopsies (i, ii, and iii, in dotted circles) from a cryosectioned tissue slice. Biopsies are mounted onto an tDISCO top-plate, fixed and stained. Cartoon of the biopsies in a brain section is shown on bottom right. Biorender was used to create the schematic. **C** Representative tDISCO-captured images of cortical brain tissue from which DAPI + NEUN+ neurons (red) and DAPI + GFAP+ astrocytes (green) were selected. Top image shows 20X field of view, scale bar is 100 μm. 100X images show NEUN+ (red dotted circles) and GFAP+ (green dotted circles) cells before (middle image) and after (bottom image) laser lysis. Scale bars are 10 μm. **D** Heatmap depicts differences in TPM normalized gene expression (columns) between NEUN+ and GFAP+ cells (rows). *n* = 4 GFAP+ and 4 NEUN+ cells from across 3 animals. **E** Heatmap depicts differences in LFQ normalized protein expression (columns) between NEUN+ and GFAP+ cells (rows). *n* = 3 GFAP+ and 3 NEUN+cells from across 3 animals. **F** Representative image of stroke-injured tissue stained for GFAP (green) from which GFAP+ astrocytes were selected. GFAP+ cells were isolated from three 200 μm "zones" (**A** through **C**) that spanned the d10 injury site (top image) and from a more distal zone (zone D, bottom image). Zone boundaries are shown by dotted lines, zone A is red, zone B is orange, zone C is light blue, and zone D is dark blue. The position of each selected astrocyte within the tissue is depicted by numbered circles. All scale bars are 100 μm. cc = corpus collosum, V = ventricle and ML = midline. **G** Transcriptome heatmap of numbered GFAP+ cells (1-22, bottom) from F (columns) isolated from zones **A** (red), **B** (orange), **C** (light blue) and **D** (dark blue). Gene expression is TPM normalized. Transcripts are grouped according to their localization in injury zone B (top), in every zone except injury zone B (second down), to distal zones **C** and **D** (third down) or lncRNA outside of zone **B** (fourth down). n = 22 cells from across 3 animals. **H** Proteome heatmap with numbered GFAP+ cells (23–32, bottom) from F (columns) isolated from zone B (orange) and zone D (dark blue). LFQ normalized intensity of proteins. Proteins are grouped according to expression within zone B (top) or in zone D (bottom). n = 10 cells from across 3 animals. Source data are provided as a Source Data file.

(Fig. 5, Supplementary Fig. 9). These candidate genes were selected to highlight examples of 1) genes that showed similar expression patterns within the infarct area (Visium) or in proximal cells (10X Chromium and tDISCO) (*Fabp5*, *Apoe*, and *Cd81*) or 2) genes that showed conflicting expression patterns in one of the three platforms (*Aldoc* and *Clu*). We validated all five of these candidates with a fourth platform, RNAScope.

We first looked at *Apoe*, *Fabp5*, and *Cd81*, genes associated with astrocytes within the infarct across all three platforms. *Apoe* was spatially restricted to the infarct site in 10X Visium sections (Fig. 5A), enriched in proximal astrocytes in 10X Chromium datasets (Fig. 5B), and identified in GFAP+ astrocytes within zone B of the infarct site tDISCO (Fig. 5C). RNAScope confirmed the presence of a spatially restricted population of *Apoe*-expressing GFAP+ astrocytes within the infarct site (Fig. 5D). Similar results were seen with *Cd81* (Supplementary Fig. 9A–D) and *Fabp5* (Supplementary Fig. 9E–H).

RNAScope validated the cross-platform analysis and confirmed that astrocytes contribute to an *Apoe*, *Cd81*, and *Fabp5*-rich signature within the stroke infarct site. Of interest, when the spatial localization of these genes was compared within Visium and RNAScope datasets, slightly different patterns or levels of expression were observed. In the Visium data, *Cd81* and Fabp5 showed less gene expression compared to *Apoe* within the infarct site (Fig. 5A and Supplementary Fig. 9A, E). Similarly, RNAScope analysis showed less intense staining of *Cd81* compared to both *Fapb5* and *Apoe* (Fig. 5D and Supplementary Fig. 9D, H).

Next, we examined two candidates that showed different trends across the three platforms (*Clu* and *Aldoc*). *Clu* was absent from the injured area in the d10 Visium dataset (Fig. 5E). However, *Clu* was upregulated in proximal astrocytes (Fig. 5F) and proximal GFAP+ zone B cells (Fig. 5G) in the 10X and tDISCO datasets, respectively. Validation with RNAScope showed expression of *Clu* in GFAP+ cells throughout the injured hemisphere. However, no differences in proximal versus distal astrocytes were seen (Fig. 5H). These detection differences could be due to the expression of *Clu* in a few (high expressing) proximal astrocytes or the false capture of *Clu* expression in both 10X Chromium and tDISCO platforms. *Aldoc* showed a similar discordance across datasets. *Aldoc* was absent in the infarct area in Visium sections (Supplementary Fig. 9I) but present in Chromium and tDISCO datasets (Supplementary Fig. 9J, K). RNAScope showed *Aldoc* expression in all astrocytes in both injured animals and uninjured controls (Supplementary Fig. 9L). This suggests that astrocytes may not differentially express *Aldoc* after injury and is in line with previous work that suggests *Aldoc* as a reliable pan-astrocyte marker[2].

Altogether, these data showcase the strengths and weaknesses of the different platforms. The platforms converged to identify robust gene targets, but they highlighted how looking at the results from a single platform could lead to misinterpretations of gene enrichment in the injured cortex. The genes *Apoe*, *Cd81*, and *Fabp5* showed complementarity in enrichment in the stroke infarct site in the subacute stage across all four platforms. The genes *Clu* and *Aldoc* were not detected by Visium within the stroke infarct, likely due to a low contribution of *Clu* and *Aldoc* to the average gene expression per spot. In contrast, the single-cell granularity offered by 10X Chromium and tDISCO, captured the astrocytic expression of *Clu* and *Aldoc* within the infarct area, but RNAScope did not detect upregulation of these genes in proximal astrocytes.

## Discussion

Here we have described how the molecular signatures of stroke were interrogated across the three platforms: Visium, 10X Chromium, and tDISCO. This breadth of analysis highlighted the importance of spatial context for -omics data from Visium and tDISCO and emphasized the need for single-cell granularity offered by 10X Chromium and tDISCO. Ultimately, we combined the high-throughput single-cell ability of 10X Chromium with spatially resolved Visium and improved our resolution of the astrocytes in the stroke-injured brain with tDISCO, our spatially resolved single-cell -omics platform.

With Visium, we obtained broad molecular signatures of stroke in time (acute, subacute, and chronic) and space and found that glial-related signatures were more pronounced in subacute stroke (d10). Next, we performed a more focused analysis of astrocytes in subacute stroke with 10x Chromium, which revealed the breadth and heterogeneity of astrocyte gene expression in the injured cortex. By integrating 10x Chromium and Visium (overlaying scRNA-seq data with Visium section), we uncovered two putative astrocyte populations, distal and proximal to the lesion site. tDISCO was used to specifically select astrocytes at desired locations around the stroke lesion site and provide scRNA-seq and scProteomics data. tDISCO allowed us to directly demarcate the spatial boundary between molecularly distinct astrocytes by isolating individual astrocytes at defined distances from the stroke lesion site. By analyzing both transcriptomes and proteomes in spatially defined subsets (proximal zones A-B and distal C-D), we found that zone B proximal astrocytes reside in a border zone, 100-200uM from the injury, and may be involved in synaptic preservation, determined by their enriched expression in neurotrophin[31] and MAPK pathways[32]. In addition, these zone B proximal cells express genes and proteins related to lipid shuttling and energy metabolism.

As expected, these three platforms (Visum, 10X Chromium, and tDISCO) produced signatures that were sometimes conflicting, likely due to the gene detection methods. These capture methods include capturing the average gene expression signal of several cells in a given space (Visium), capturing a single cell with no spatial context

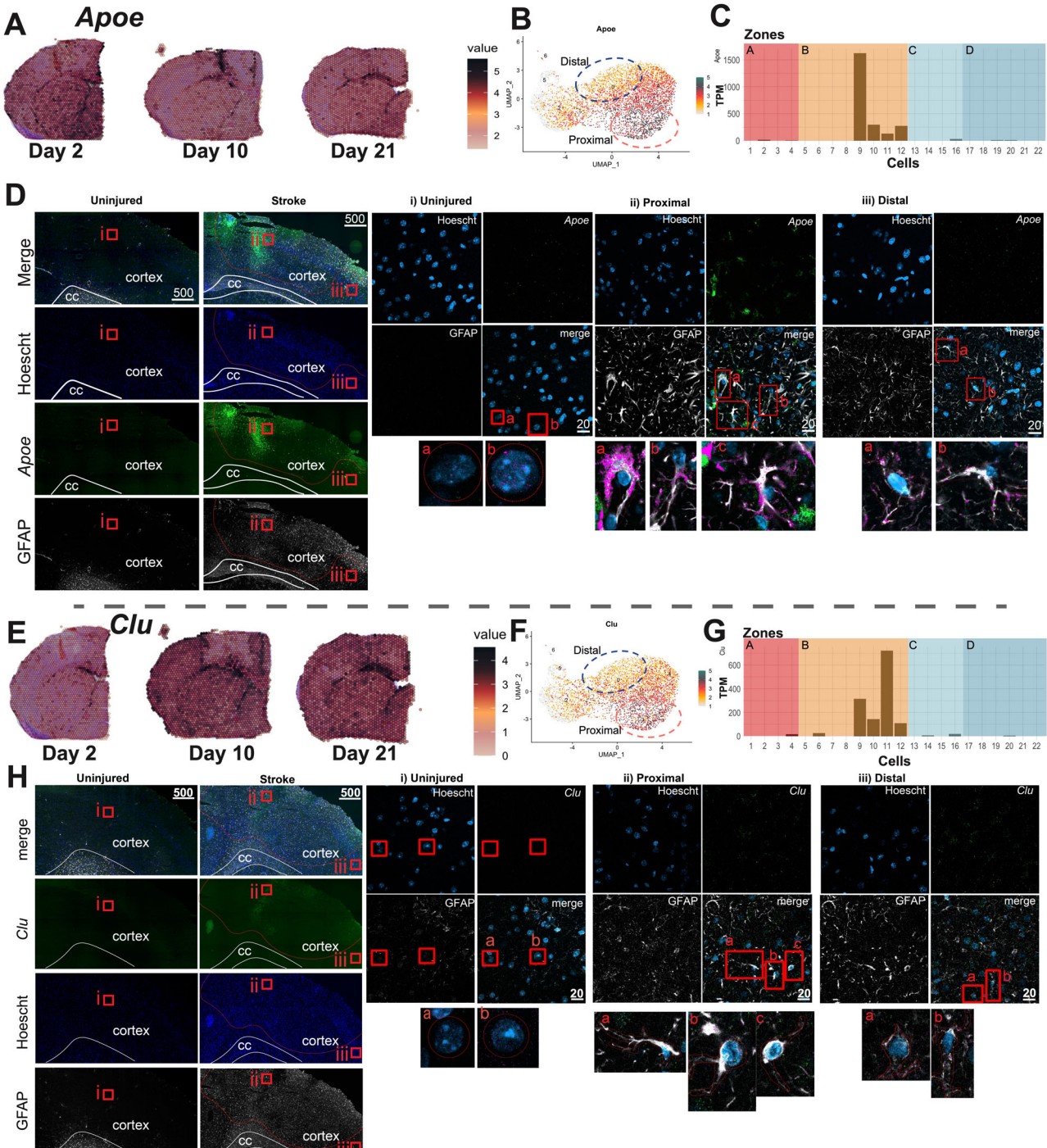

**Fig. 5 | The expression of *Apoe* and *Clu* across platforms and validation with RNAScope. A** Visium d2, d10 and d21 sections show high *Apoe* gene expression that is specific to the d10 injury site. **B** UMAP plot of the 10X Chromium scRNA-seq data annotated with cluster 3 (peach, proximal astrocytes) and cluster 0 (navy blue, distal astrocytes) shows *Apoe* expression is localized to cluster 3 (proximal) astrocytes. **C** Histogram depicts tDISCO-derived TPM normalized gene expression (y-axis) of *Apoe* in GFAP+ cells (numbered 1-22, x-axis) from zones **A** (red), **B** (orange), **C** (light blue), and **D** (dark blue). **D** Representative images of *Apoe* and GFAP expression in uninjured (left column) and d10 stroke-injured brains (second from left column). The red dotted line in stroke-injured brains denotes the injured area as defined by GFAP staining. Red boxes show the location of the higher

magnification images that correspond to the uninjured brain (i), and proximal (ii)- and distal (iii)- regions in the stroke-injured brain. Images are at 10x, scale bar = 500 μm. Higher magnification images show *Apoe* and GFAP expression in i) uninjured, (ii) proximal and (iii) distal astrocytes (colocalization is depicted by the pink overlap mask). Images are at 40X, scale bars = 20 μm, n = 3 mice. cc = corpus callosum. **E-H** *Clu* expression in Visium (**E**), 10X Chromium (**F**), and tDISCO (**G**) datasets. Representative images of *Clu* and GFAP expression in uninjured (left column) and d10 stroke injured brains (second from left column), and in i) uninjured, (ii) proximal and (iii) distal GFAP+ astrocytes (colocalization is depicted by pink overlap mask) (**H**). Images are at 40X, scale bars = 20 μm, n = 3 mice. cc = corpus callosum. Source data are provided as a Source Data file.

(10X Chromium), and capturing individual cells with several appended selection criteria, such as spatial context, neighboring cell composition, cell marker staining information, distance to injury (tDISCO). Genes appearing in all analyses and some with conflicting expression patterns across platforms were validated with RNAScope.

The genes *Apoe*, *Cd81*, and *Fabp5* were validated to be present in proximal astrocytes. This is in line with a previous study of subacute stroke where two distinct populations of APOE-expressing astrocytes were identified: infarct-associated APOE + GFAP+ cells that may correspond to our proximal cells and APOE + S100$\beta$+ astrocytes that formed a peri-infarct boundary[33]. In terms of function, *Apoe* and *Fabp5* are known to coordinate lipid metabolism between astrocytes and neurons[19]. Alterations in both of these genes are associated with neurodegenerative pathologies[34,35]. In stroke, *Apoe* knockout mice were shown to have larger infarct sizes[36] in the acute stage, while in subacute stroke APOE reduction is associated with improved recovery[33]. Altogether, this suggests that *Apoe* and *Fabp5* play important roles in astrocytes to influence stroke recovery.

With RNAScope, we validated that *Clu* and *Aldoc* were expressed in astrocytes in the injured cortex (although not enriched in proximal or distal cells). This highlights the mechanism by which Visium mainly captures the highest gene expression averaged from several cells and cell types, and therefore missed nuanced differences in astrocyte gene expression and failed to capture the expression of *Clu* or *Aldoc* in the stroke-injured cortex. 10X Chromium captured astrocytic-specific gene expression, including *Clu* and *Aldoc*, but spatial context could only be inferred indirectly by integration with Visium. In contrast, tDISCO was able to place these *Clu* and *Aldoc* expressing astrocytes in the Zone B injury space. These two genes highlight the importance of single-cell granularity, while maintaining spatial context information.

We propose that tDISCO fits a cavity in the emerging trend of spatially resolved single-cell sequencing techniques. As illustrated here, the capacity to select individual cells provides a complementary advantage to Visium, which does not provide the same resolution and is limited to NGS. Furthermore, there are lower cell input requirements for tDISCO, which can ultimately drive down the cost of a scRNA-seq experiment. Examples of other platforms capable of similar cell selection include the cutting-edge Image-seq technique[37], which provides similar capabilities in true in vivo specimens, but requires precise manipulation of a micropipette for aspiration at each step, reducing throughput. Similarly, Niche-seq[38] allows high-resolution, spatial labeling of cells, but requires FACS to isolate these individual cells for sequencing. Lastly, High Definition Spatial Transcriptomics (HDST)[39] and Slide-seq[40] provide a high-resolution version of a method similar to Visium but is also limited to NGS of an entire barcode array overlaying a tissue, thus lacking the selection criteria offered by tDISCO. In contrast to the above, tDISCO is fully integrated in a miniaturized device with no moving parts, and is method-agnostic, giving the user the flexibility to choose which -Omics (e.g., NGS, proteomics) to survey on select cells. In its current form, tDISCO was used primarily for astrocyte selection and isolation, with processing for NGS and proteomics carried out off-chip. In the future, however, tDISCO could be integrated with state-of-the-art automated digital microfluidic pipelines for NGS[41,42] and proteomics[43,44] sample processing, providing a hands-free user experience from sample-to-sequence. Further miniaturization and integration may also improve sensitivity and reduce the time-to-answer, as has been observed in the integration and miniaturization of other neurodiagnostic assays[45].

In summary, we present a platform for spatially resolved -omics and demonstrate how different technologies can resolve cellular heterogeneity after injury through the integration of spatial information.

## Methods
All animal work complied with regulations in Canada and was approved by the Animal Care Committees at the Toronto Center for Phenogenomics at the Lunenfeld-Tanenbaum Research Institute at Mount Sinai Hospital and the Department of Comparative Medicine at the University of Toronto.

### Mice
8–10-week-old male C57/Bl6J mice purchased from Charles River were used for all experiments. No analysis of sex was carried out in this first study. Experiments were conducted according to protocols approved by the Toronto Centre for Phenogenomics (Animal Use Protocol #26-0392H) and the Department of Comparative Medicine of the University of Toronto (Animal Use Protocol #26-0392H)01155335). 8–10-week-old B6JcFVB/N-Tg (GFAP-GFP)14Mes/J (original FVB Strain: 003257 from The Jackson Laboratory) mice were also used for the RNAScope experiments wherein the native fluorescence signal from GFP was quenched by the overnight treatment with 70% ethanol used in the staining protocol.

### Endothelin 1 (ET-1) stroke
Cortical ischemic lesions were produced by injecting 1 μL 400 picomolar of ET-1 (R&D Systems, Cat: 1160) dissolved in filtered saline (1 μg/μL) into the right forelimb sensory-motor cortex at the coordinates 0.6AP, 2.25 ML, and −0.1DV from bregma[4]. A 29 gauge Hamilton syringe with a 45 degree beveled tip was used to inject ET-1 at a rate of 0.1uL/min. The needle was left in place for 10 minutes after ET-1 injection to prevent backflow and then slowly withdrawn. Sham mice received saline injections.

### Visium spatial transcriptomics
Brains were snap-frozen and sectioned according to 10x Visium protocols. Initially, two tissue sections from one mouse, at the coordinates AP 0.5 from bregma were collected for each group (day 2-, day 10-, and day 21-stroke injured, and a day 2 sham control). In a second experiment, two more sections for each time point were collected, totaling 3 biological replicates per time point. Slides were then fixed (methanol, as per Visium guidelines), permeabilized (18 min), processed and sequenced (Novaseq6000, 100 cycles) at the Princess Margaret Genomics Center at the University Health Network (Toronto, Canada).

The data for each section was aligned to the mouse reference genome (GRCm38/mm10) using Space Ranger 1.0.0. All data analysis and visualization was done using STUtility (v0.1.0)[46]. Filters on Visium spots include a min.gene.count of 100 and min.gene.spots of 5. All images were rotated and adjusted to be visualized in the same direction using WarpImages and negative binomial regression was used to normalize the UMI count data using SCTranscform. Data was first clustered using non-negative matrix factorization (NMF) to obtain genes driving less spatially discrete clustering offered by NMF, using RunNMF with 20 factors. Data was then clustered using RunUMAP, using 20 dimensions and n.neighbors = 10. Neighbor analysis in Fig. 2E–G was done using RegionNeighbours. Metrics relating to read, UMI gene detection and mitochondrial and ribosomal content are in Supplementary Fig. 10. For the Deconvolution of cell types using the Allen Brain Atlas as a reference (Supplementary Fig. 2), *TransferData* from Seurat (v4.2.0) was used to overlay reference data provided by the Allen Brain institute (*allen_cortex.rds*)[47], using the same spot gene count genes per spot parameters used in STUtility analyses (above).

### 10X Chromium sc-RNAseq cell isolation
Two independent experiments were performed. In the first experiment (data in Fig. 3A–C), cells were collected from a biological pool of *n* = 12 mice from stroke-injured or uninjured controls. Mice were euthanized by cervical dislocation under isoflurane anesthesia. Brains were harvested and washed in Ca$^{2+}$/Mg$^{2+}$ free PBS supplemented with DNaseI (Sigma-Aldrich, D4527, 0.025 mg/mL) and RNaseOUT recombinant ribonuclease inhibitor (ThermoFisher Scientific, 10777019 1:40000) referred to as PBS-RID. Right sensorimotor cortices were

micro-dissected with a scalpel blade and pooled for digestion using the Adult Brain Dissociation Kit (Miltenyi Biotec, 130-107-677) and octo-MACS settings from Liu et al., 2021[48]. The resulting single cell suspension was incubated in FcR blocking reagent (Miltenyi Biotec, 130-092-575) in AstroMACS Separation Buffer (Miltenyi Biotec, 130-117-336) for 15 minutes at 4 ˚C. Cells were then stained with ACSA2-microbeads (Miltenyi Biotec, 130-097-679, 1:10) and ACSA2-PE (Miltenyi Biotec, 130-116-244, 1:100, clone: IH3-18A3) for 15 minutes at 4 ˚C. Cells were washed with AstroMACS Separation Buffer (Miltenyi Biotec, 130-117-336) and MACS was performed using an MS column (Miltenyi Biotec, 130-042-201) according to manufacturer specifications (Miltenyi Biotec, 130-097-679). The ACSA2+ fraction was washed with PBS-RID, and the cells were resuspended in PBS-RID for FACS sorting. Prior to FACS sorting cells were stained with DRAQ5 (Invitrogen) and DAPI (Invitrogen). The stained cells were sorted on a Beckman Coulter MoFlo Astrios EQ cell sorter (Lunenfeld-Tanenbaum Research Institute, Toronto, Canada) purifying for DRAQ5 + DAPI-ACSA2PE+ cells for downstream 10X scRNA-seq.

In the second experiment (data in Fig. 3D–I), cells were collected from a biological pool of $n = 8$ mice from stroke-injured or uninjured controls. Mice were euthanized by cervical dislocation under isoflurane anesthesia. Brains were harvested and washed in cold PBS and the right sensorimotor cortex was micro-dissected. Tissue from each experimental condition was pooled and dissociated using the Neural Tissue Dissociation Kit (P) (Miltenyi Biotec, 130-092-628). Following dissociation, myelin was removed using the Myelin Beads Removal II Kit (Miltenyi Biotec, 130-096-733) with an MS magnetic column (Miltenyi Biotec, 130-042-201). The resulting cell suspension was incubated with FCR blocking reagent (Miltenyi Biotec, 130-092-575) for 10 minutes at 4 ˚C and then incubated with anti-mouse ACSA2-PE (Miltenyi Biotec, 130-116-244, 1:100) for 15 minutes at 4 ˚C. To isolate ACSA2-PE+ cells, MACS was performed using anti-PE microbeads (Miltenyi Biotec, 130-048-801) with an MS magnetic column (Miltenyi Biotec, 130-042-201) according to manufacturer specifications. The ACSA2-PE+ fraction was collected and resuspended in FACS buffer (1x PBS supplemented with 0.5% BSA and 2 mM EDTA), stained with DRAQ5 (Invitrogen) and DAPI (Invitrogen) prior to sorting. Live, nucleated, PE + FACS sorted cells were collected for 10X scRNA-seq. 10X Chromium was performed and then sequenced (Novaseq6000, 100 cycles) at Princess Margaret Genomics Center at the University Health Network (Toronto, Canada).

The data for each group (uninjured and stroke samples) were filtered for low-quality reads and then aligned to the mouse reference genome (GRCm38/mm10) using CellRanger. Cells were sorted according to the barcodes and the unique molecular identifiers (UMIs) were counted per gene for each cell. Metrics relating to read and gene detection and mitochondrial content can be found in Supplementary Fig. 11.

Gene-barcode count matrices were analyzed with the Seurat R package (version 4.0.6)[49]. In total, 3092 and 2603 cells as well as 14784 and 14899 genes were captured for uninjured and stroke libraries, respectively. To remove possible doublets, we removed the cells with more than 3800 total expressed genes. Cells with less than 200 genes detected and more than 30% mitochondrial gene mapped reads were removed from downstream analyses (Supplementary Fig. 11). After this procedure, 3009 and 2529 cells remained in uninjured and stroke groups, respectively. The merged stroke and uninjured Seurat object was log-normalized and scaled. For dimensionality reduction, the most variable features/genes were determined using the FindVariableFeatures. Dimensionality reduction was then performed using PCA, and UMAP plots were generated by the RunUMAP Seurat function with the first 10 PCs as input. We identified differentially expressed genes in the specific cluster when compared to other remaining clusters with the FindAllMarkers Seurat function. A marker gene was defined as showing >0.25 log-fold expression change compared to the mean expression value in the other clusters, and with a detectable expression in >25% of all cells from the corresponding cluster, and adjusted P-value < 0.05. The DoHeatMap function was used to plot the top 10 overexpressed marker genes for each cluster. Once marker gene signatures of each cluster were identified, gene ontology enrichment analysis (GOEA, http://www.geneontology.org) 6–8 and gene set enrichment analysis (GSEA) 9 were carried out to evaluate alterations of hallmark biological processes and pathways related to each cell cluster. For integrating the 10x scRNAseq data and day 10 Visium data we used annotations driven from scRNA-seq (as a reference) were transferred to the Visium data (as the query), using the anchor-based integration workflow[50]. The procedure outputs a probabilistic a prediction score for each cell in the query dataset, which is added as a new assay in the Seurat object. The SpatialFeaturePlot function in Seurat was used to visualize the probabilistic localization of sc-RNA-driven clusters on the Visium dataset.

**tDISCO Instrument assembly**

The tDISCO instrument (Fig. 4A) is based upon an upright modular microscope system (Applied Scientific Instrumentation [ASI], RAMM/MIM system). Briefly, from the sample plane upwards: infinite conjugate objectives (Olympus UPlanSApo 4x, Olympus LUCPlanFL N 20x, Nikon CFI Plan LWD IMSI 100X) are connected to a motorized turret (ASI); an IR beamsplitter interfaces with the CRISP autofocus module (ASI); a motorized filter cube slider (ASI) allows switching between a 4-channel full multiband filter set (Semrock, LED-DA/FI/TR/Cy5-B-000) for fluorescence imaging and an empty position for brightfield and laser lysis; an 532 nm reflecting band beamsplitter (Chroma Technology Corp., ZT532dcrb) couples the lysis beam into the objective path; finally, a 200 mm tube lens (ASI) projects the objective image onto an 8.9 megapixel monochrome CMOS camera (Thorlabs Inc., CS895MU). These components are mounted on the LS-50 focus drive backbone (ASI) allowing focus by adjusting the distance between the objective turret and sample stage. The fluorescence excitation is provided by a multichannel LED/laser source (Excelitas Technologies, X-Cite TURBO), allowing the visualization of different fluorophores with a single filter set by selecting different excitation sources. The lysis laser beam is produced by a sub-ns Q-switched Nd:YAG laser (Passat Ltd., Sub-Naples Mini 532 nm), interfaced with the microscope using two multimode optical fibers, with a variable attenuator between the fibers allowing reduction of the laser power (Thorlabs, VOAMMF). Below the motorized sample stage (ASI), a condenser module and green LED source (ASI) provide illumination for brightfield imaging. Plugins to the open-source DropBot[51] control system were written to allow for integrated programming and control of automated microscope objective selection and focus, fluorescent and bright field illumination, pulsed laser lysis, and droplet control.

Devices used with tDISCO are similar to those used in the original DISCO[7] method, with the critical difference of being designed to interface with tissue slices instead of in vitro cultures. Briefly, day 10 stroke injured brains ($n = 3$) brains were snap frozen in an isopentane dip and stored in the −80 °C. Prior to sectioning, tissues were equilibrated to the −20 °C cryosection chamber for at least one hour. A biopsy punch was used to demarcate a 2 mm diameter region of interest (ROI) (Supplementary Fig. 6 & Fig. 4B). Tissues were cryosectioned at 10 μm at −20 °C. The tissue around the demarcated ROI was removed and the ROI was mounted onto the corresponding 2 mm hydrophilic spots of the DISCO top-plate (Supplementary Fig. 6 & Fig. 4B), these hydrophilic spots were coated with poly-L-lysine (Sigma-Aldrich, P4707). The DISCO top-plated mounted tissues were then fixed in 80% methanol for 30 min at −20 °C, blocked for 10 min (3% SSC buffer + 2% BSA + 0.1% triton X-100 + 1U RNAse inhibitor (Lucigen, MA125)), then stained against GFAP (1:100, AvesLabs, AB_2313547, polyclonal), NeuN (1:100, EMD Millipore, ABN78, polyclonal)) and DAPI (1:10000, Invitrogen, D3571), using secondary antibodies at a 1:100

dilution as well (Invitrogen, A11039); primary and secondary antibodies were used at a volume of 1 μl per tDISCO tissue section (Goat anti-chicken IgY, Alexafluor 488, Invitrogen, A11039, polyclonal, 1:100; Goat anti Rabbit IgG Alexafluor 647, Invitrogen, A32733, polyclonal, 1:100). All wash steps and final incubation were done with the same buffer: 1X SSC buffer, 0.001% T90R4, 1U RNAse inhibitor (Lucigen, MA125). Primary antibodies were validated in negative (1a absent) and positive control tissue and optimized using serial dilution. These antibodies have also been extensively validated by manufacturers showing specificity towards these proteins in the brain for our application of choice (immunohistochemistry or MACS). A list of references citing the usage of this antibody are found on the manufacturer's website.

Top-plates were then assembled with bottom plates and interfaced with the tDISCO system via a new magnetic clamping mechanism (that allows for rapid exchange of devices). After laser-lysis of a cell (in some cases, one cell lysate per droplet; in other cases, multiple cell lysates per droplet), 3 μL droplets of 1X SSC buffer, 0.001% T90R4, 1U RNAse inhibitor (Lucigen, MA125) were used to collect/wash the cell lysate and prepare the section for another round of laser-lysis. Lysis mechanism was similar to what was reported previously[7], except with the application of 3 laser pulses per cell at a frequency of 800 Hz. After experiments were concluded, droplets containing cell lysates were collected into separate tubes and held at −80 °C until further processing for transcriptomics or proteomics.

## tDISCO transcriptomics
Cell lysates that were snap frozen were brought through the reverse transcription (RT), first strand synthesis. First, cell lysates were incubated with custom cell barcodes (25 μM, Supplementary Table 1), 1X lysis buffer (Takara, 635013) and dNTPs (Thermofisher, R1121, 25 μM of each) for 5 min at 72 °C, then ice. The remaining RT reagents: Maxima H Reverse Transcript minus (ThermoFisher, EP0751), 5X buffer, RNAseq Inhibitor (Lucigen, MA125) and a custom template switching oligo (TSO, 10 μM, Supplementary Table 1) were then added to the cell lysates and incubated for 1.5 hours at 42 °C. Cell lysates in groups of 10 were then pooled and brought through second strand synthesis and cDNA amplification using further indexing custom primers (10 μM, five of them in Supplementary Table 1) and SeqAmp DNA polymerase (Takara, 638509). Finally, cell lysates were further pooled to 50 cells per Illumina index and brought through the standard Nextera XT prep using a custom P5 oligo and standard N714 and N715 Illumina indices. A library of 50 cells was then run using a Miseq V3 kit 75 × 75, with a custom Read 1 and custom Index 2 primer, with cycle parameters of 36 bp for Read1 (Cell barcode and Second strand synthesis barcode), 8 bp for Index2 and 127 bp for Read2 (cDNA content). An Illumina provided custom script was also used on the Miseq (provided on github.com/eyscott) to enable 3 forward reads and the use of 2 forward reading indices. Read and gene count and gene type composition metrics for these libraries can be found in Supplementary Fig. 12. The Fastq files obtained were then trimmed (fastp, v0.20.1)[52], aligned to GRCm38 (StarAligner, v 2.7.9)[53], demultiplexed and UMIs collapsed using custom scripts, similar to those published in the previous DISCO manuscript[7]. Count tables were then collected using featureCounts from Rsubread (v2.10.5)[54], normalized to transcripts per million and cells were matched back to their tDISCO obtained images (which includes spatial proximity to the stroke injection site and cell staining profile). Differential gene expression analyses were done using likelihood ratio tests provided by edgeR (v3.38.4)[55] and final data visualization was done using ggplot2 (v3.3.6)[56] and gplots (v3.1.3)[57] in R (v4.2.0).

## tDISCO proteomics
The proteomic analysis was performed off-chip (in capped tubes) and similar to what was previously done[7]: a 4 μL droplet (1X PBS + 0.0025%

w/v n-Dodecyl β-D-maltoside (DDM) containing cell lysate was reduced in 5 mM tris(2-carboxyethyl)phosphine (TCEP, final concentration) at 70 °C for 30 min (thermocycler, Bio-Rad), alkylated with 10 mM iodoacetamide (IAA, final concentration) in the dark at room temperature for 30 min, then digested at 37 °C with 0.5 μL Lys-C (5 ng/μL) for 3 h and 0.5 μL of trypsin (20 ng/μL) for 16 h sequentially. The product was quenched with 0.2 μL of formic acid before nano liquid chromatography-mass spectrometry (nanoLC-MS) analysis.

An EASY-nLC 1200 ultra-high-pressure system coupled to a Q Exactive HF-X mass spectrometer equipped with a nano-electrospray ion source (Thermo Fisher Scientific) was used to perform the nanoflow reversed-phase LC. The prepared peptide samples were loaded onto a 12 cm fused silica microcapillary column (100-μm i.d., Polymicro Technologies), packed in-house with 1.9 μm ReproSil-Pur C18 120 Å reversed phase particles (Dr. Maisch GmbH). Mobile phase A (water with 0.1% formic acid, v/v) and mobile phase B (80/20/0.1% ACN/water/formic acid, v/v/v) were applied to separate the peptides with a constant flow rate of 300 nL/min with a linear gradient of 3–30% mobile phase B within 90 min, followed by a linear increase from 30–45% mobile phase B within 20 min, then a linear increase from 45–95% within 1 min and a 14-min plateau before re-equilibration. Known quantities of HeLa standards were run in triplicate to assess the sensitivity and consistency of this pipeline (Supplementary Fig. 13A), which a single blank was also alongside our single cell lysate samples to show low contamination (Supplementary Fig. 13B).

Standard shotgun LC-MS experiments were performed with a data-dependent top10 method. A full MS scan range of m/z 375–1575 at a resolution of 120,000 at m/z 200 with an AGC target of $5 \times 10^5$ ions and a maximum injection time of 50 ms. Precursor ions with charges of +2 to +6 were isolated with an m/z window of 2 and fragmented by high energy dissociation with a collision energy of 27% at a resolution of 60,000 at m/z 200. MS/MS scans were performed in the Orbitrap with the AGC target, and injection time set to $5 \times 10^4$, 250 ms respectively. Previously targeted precursors were excluded from re-sequencing for 20 s. Raw files were analyzed by MaxQuant (version 1.6.4.0)[58]. MS/MS spectra were searched against human protein database from Uniprot, allowing for variable modifications of methionine oxidation and N-terminal acetylation and fixed cysteine carbamidomethylation. The specific proteolytic enzyme was trypsin. The minimum peptide length was six amino acids and the maximum peptide mass was 4600 Da. The allowed missed cleavages for each peptide were 2. Matching between runs (MBR) was applied in MaxQuant while performing label-free quantitation (LFQ). Both peptides and proteins were filtered with a maximum FDR of 0.01. The other parameters were set as default. Differential enrichment analysis of proteins was determined using the package DEP (v1.18.0)[59], using the test_diff() function, which performs differential protein analysis by combining linear models with empirical Bayes statistics using limma. Parameters used for test_diff() were alpha = 0.05 and lfc = 1, and data was visualized using heatmaps provided by gplots (v3.1.3)[57].

## Single-molecule FISH
Snap frozen coronal sections were collected from uninjured and day 10 post ischemic stroke brains and stained using RNAScope smFISH probes and detected using RNAscope Multiplex Fluorescent Assay Kit (all from Advanced Cell Diagnostics, ACD). Briefly, sections were fixed in chilled 4% paraformaldehyde (PFA) for 15 min at 4 °C, washed twice in 1X PBS, and permeabilized in 50 μL of 70% ethanol overnight at 4 °C beneath a parafilm square. The following day, sections were washed twice in 50 μL of 2X saline sodium citrate buffer (SSC) for 10 min. smFISH Probes (Advanced Cell Diagnostics) were pre-heated for 10 minutes at 40 °C, cooled to room temperature (RT) and added to sections to incubate (always in a humidified chamber) for 2 h at 40 °C. Probes were used to target *Apoe* (ACD Cat No. 313271,

NM_009696.3), *Fabp5* (ACD Cat No. 504331, NM_010634.3), *Cd81* (ACD Cat No. 556971-C2, NM_133655.2), *Clu* (ACD Cat No. 427891-C2, NM_013492.2), and *Aldoc* (ACD Cat No. 429531-C3, NM_009657.3) mRNAs. Following probe incubation, sections were washed (always with RNAScope wash buffer for 1 min) four times, incubated in RNAscope AMP-1 solution for 30 minutes at 40 °C, washed four times, incubated in RNAscope AMP-2 solution for 15 min at 40 °C, washed four times, incubated in RNAscope AMP-3 solution for 30 min at 40 °C, washed four times, incubated in RNAscope AMP-4 solution for 15 min at 40 °C and washed four times. For concomitant immunostaining, sections were incubated in 5% BSA blocking buffer for 1 hr at RT, washed once with 1X PBS, and incubated in primary antibody solution [rabbit anti-GFAP (DAKO Dako Z0334, polyclonal) diluted 1:5000 in 2.5% BSA] overnight at 4 °C. The following day, sections were washed three times with 1X PBS for 5 min each and incubated in fluorescently labeled secondary antibody solution [donkey anti-rabbit Alexa Fluor 647 (Jackson Immunoresearch, AB_2492288, polyclonal) diluted 1:500 in 1X PBS; Jackson Immunoresearch] in the dark for 2 h at RT. Sections were washed three times in 1X PBS for 5 min, incubated in 0.5 mg/mL Hoechst 33258 (Sigma-Aldrich) for 2 min at RT, washed once with 1X PBS for 5 min, and mounted on glass slides using PermaFluor (Thermo Fisher). Sections were imaged on a confocal (Zeiss LSM 900).

## Reporting summary

Further information on research design is available in the Nature Portfolio Reporting Summary linked to this article.

## Data availability

The sequencing and proteomic data generated in this study have been deposited in the NCBI SRA database under Bioproject number PRJNA952594 and in ProteomeXchange under accession code PXD041388. The processed sequencing and proteomics data, as well as the code used to process them are available at https://github.com/eyscott/Visium_10XChromium_tDISCO (https://doi.org/10.5072/zenodo.20771). The source data generated in this study are provided in the Supplementary Information/Source Data file. Source data are provided with this paper.

## Code availability

All custom codes used to collect data are available at http://microfluidics.utoronto.ca/gitlab/DISCO. The codes for analysis are available at https://github.com/eyscott/Visium_10XChromium_tDISCO.

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

## Acknowledgements

The authors acknowledge the Princess Margaret Genomics Center, the Donnelly Sequencing Center and Lisa Lock from Illumina. This work was supported by grants from the Canadian Institutes of Health Research (CIHR PJT-175254 to MF and SJT and CIHR PJT-175137 to SAY), the National Science and Engineering Research Council (RGPIN-2020-06512 to MF; RGPIN 2019-04867 to ARW, RGPIN-2021-03514 to SAY; RGPIN-2020-05834 to SJT), the Temerty Foundation (Pathway Grants to MF & SAY), Medicine by Design Cycle 2 Funding (MbDC2-2019-04 to MF), the Krembil Foundation (to SJT), the Canada Research Chairs Program (950-231616 to ARW), the Canadian Foundation for Innovation and the Province of Ontario (36661 to ARW), and Genome Canada/the Ontario Genomics Institute/the Province of Ontario (to ARW). ED was supported by a graduate grant from the Branch Out Neurological Foundation, a CIHR-CGS-M and an Ontario Graduate Scholarship, DLC was supported by a CIHR-CGS-M.

## Author contributions

E.Y.S., M.D., S.J.T., S.A.Y., A.W. and M.F. contributed to the conception or design of the work. E.Y.S., M.D., D.L.C., J.P., E.D., S.A., I.N.X.L., KWAB, SAY, and MF contributed to the acquisition of data. E.Y.S., N.S., T.T., S.A., J.P., D.L.C., E.D., I.N.X.L., K.W.A.B., S.J.T., S.A.Y. and M.F. contributed to the analysis of data. All authors contributed to the interpretation of data. M.D. contributed to creation of software and hardware used for tDISCO. E.Y.S. and M.F. drafted the work. S.J.T., S.A.Y. and A.W. substantively revised it. All authors contributed to the final version.

## Competing interests

M.D. and A.R.W. are co-inventors on a patent application (Applicants: The governing council of the University of Toronto, and Sinai Health System; Inventors: Michael Dryden and Aaron R Wheeler; Application Number: PCT/CA2017/051158 (filed 20.09.2019); Status: Pending; Specific Aspect of Manuscript: System and method for identifying and targeting individual cells within a cell population for selective extraction of cellular content with a pulsed laser within a digital microfluidic device.
