## [Peer Review File · Nature Communications]

Integrating single-cell and spatially resolved transcriptomic strategies to survey the astrocyte response to stroke in male mice.REVIEWER COMMENTS

Reviewer #1 (Remarks to the Author):

Here the authors use a combination of single cell RNAseq (scRNAseq), spatial transcriptomics, and a recently published laser capture method the authors label 'DISCO' to investigate the gene and protein changes in cells following stroke. Of particular note, the 'evDISCO' proteomics seems to be single cell and is a highlight of the approaches used in this study.

The authors use these tools to define Apoe and Fabp7 as differentially regulated following cortical stroke.

Overall the manuscript uses modern methods, but these are underpowered such that it is unclear if the findings are appropriate. I have a number of other concerns with the manuscript that would benefit from clarification:

1. incorrect labeling of cell types (e.g. Figure 2c) - Vimentin is a gene that is highly expressed in astrocytes and endothelial cells, while several other genes are a greater vascular gene expression signal. How was this chosen as a purely macrophage signal?

This concern also follows to cell type signatures in 'Factors' in this figure - the 'glial factor signature' is a combination of multiple cell types - presumably astrocytes (Gfap, Apoe, etc.) microglia (Aif1, Trem2, C1a, etc.), and oligodendrocytes (e.g. Plp1).

That this is so poorly defining a single cell type is problematic to the overall mechanism under investigation here.

2. overinterpretation of results. The authors make bold statements that are not currently backed up by the data that is included. An example is at line 97-101 (end of the first section) - these conclusions cannot be drawn from the data presented in this section of the manuscript. It is a hypothesis that needs to be tested, but there is no characterization of these statements. another example is in statements pertaining to the sequencing depth of 'evDISCO' - please provide details for 'higher sequencing depth' afforded using this method - if it is less than 50x that achieved by 10x chromium, the sequencing depth is actually shallower - as 50 cells have been combined for 'evDISCO' - which would place this method as a bulkRNAseq approach - which is most often deeper by virtue of a greater amount of genetic material being loaded into library preps and the sequencer. Note: the pooling of 50 cells for the 'evDISCO' transcriptomics methods seems to lower the threshold and sensitivity for these studies when compared to snRNAseq.

3. astrocyte purification/enrichment (using ACSA2 antibody) - does ACSA2 enable the collection of all astrocyte subtypes, or is it only a sample of cortical astrocytes? How does the possible sub-sampling affect the conclusions drawn from this dataset as a result? Can the authors provide evidence that GLAST and ACSA2 label the same cells in the cortex? In the cerebellum these are highlighted as labeling overlapping by different populations (see PMID: 34060113). Atp1b2, the gene encoding ACSA2 has also been reported in neuronal cells, and in non-glial and glial progenitors.

4. the only two transcripts/proteins identified as novel in this study at Apoe and Fabp5 which have been investigated with some vigor in the stroke field.

Apoe as an example was investigated by PMID: 19623195) and highlights a peri-infarct region astrocyte population that is APOE negative - the authors should discuss their results in the context of these, and other, historical data.

5. as a general statement, the new acronym 'DISCO' is poorly thought out, as this will cause confusion with iDISCO (immunolabeling-enabled three-dimensional imaging of solvent-cleared organs from the Tessier-Levigne group. In addition, the prefix of 'ex vivo' is incorrect - ex vivo refers to removal of a tissue/organ from the body for treatment and then returning it to the body. In the context of this study, ex vivo is the incorrect term

6. can the authors comment on the alarming loss of Apoe in subcortical regions >2 days following stroke? This seems to be the most striking change in expression that is maintained for at least 3 weeks.

7. underpowered studies - this is a major concern, as the validation steps are minimal (2 genes) and therefore the scRNAseq and spatial data for the basis of most of this manuscript.

scRNAseq: the cell numbers collected for scRNAseq studies are incredibly low: 3009 (uninjured, from 8 brains, ~376 cells/brain) and 2529 (stroke, from 8 brains, ~316 cells/brain).

spatial transcriptomics: only a single section is completed for each group of animals - giving an n of 1.

These both seems very underpowered and are thus unlikely to be sufficiently able to address the specific questions the authors are attempting to

Minor points:

1. Figure 2, panel C - (in relation to color scale) What is 'value' referring to? Raw expression values? Normalized values?

2. Figures (in general) - the manuscript and figures are difficult to follow (e.g. line 222 'Figure 4G, third group down'). The authors might consider a way to more clearly define these figures - perhaps labeling the individual 'groups' running down the second panel included in G and H (for example). This is a concern for many of the figures.

Panel A in Figure 4 offers considerably little detail about the 'evDISCO' method - as a reviewer I had to read the recent manuscript from Lamanna in Nat Comm to understand what exactly had been done.

3. Figure 5, panels E-H - what time following stroke are these images take?

Reviewer #2 (Remarks to the Author):

The authors developed a platform called evDISCO for spatially resolved single-cell transcriptomics and proteomics. In order to examine astrocyte diversity in stroke, they utilized evDISCO alongside two high-throughput platforms for spatial (Visium) and single-cell transcriptomics (10x Chromium). Integrating Visium and 10X Chromium datasets, they identified two astrocyte populations, either proximal or distal to the injury site. Then, evDISCO enabled further refinement of the spatial boundaries that define these populations and revealed their molecular profiles. This study provided a significant resource for elucidating the roles of astrocytes in stroke. However, the article requires some revisions.

1) It would be helping to provide a more comprehensive explanation for the selection of the time points 2, 10, and 21 days.

2) The article has a number of minor errors. For instance, the illustration in Supplemental Figure 2 shows there are three rows, but only the third row is visible. And reference 4 is a 2020-accepted article; therefore, a volume and issue number must be provided.

3) For the conclusion of Figure 3B, namely that the proportion of each cell subtype differs between stroke and control, is there any way to determine if this is due to randomness, given that there appears to be only one sample for stroke and control?

4) How can the influence of chance on the conclusions be excluded when there appears to be a single sample for all three types of data? As the authors note later, there are also inconsistencies between the three types of data; are these inconsistencies the result of randomness?

5) How were the genes Fabp5, Apoe, and Cd81 identified from the 1,221 differentially expressed genes?

6) The pixels in Supplemental Figure 4 are too small and do not appear to effectively convey the most important platform characteristics and optimization information.

7) The article fails to introduce and explain the distinctions between zones A, B, C&D, despite the fact that zone A is closer to the injury site than zone B. Could this be discussed or explained? In addition, is there any functional overlap or similar enrichment between the differential expression of proximal and distal astrocytes identified by evDISCO and the differential expression identified by

10x Chromium scRNA-seq data?

8) According to the 10x Chromium scRNA-seq data, the genes *ApoE* and *Fabp5* are enriched in astrocytes proximal to the lesion site. The authors present the expression of mRNA in the evDISCO data; are expression data for proteins available?

9) The description of Figure 5 in "Results" is insufficient. The results in the "ApoE and Fabp5 are enriched in astrocytes proximal to the lesion site" are inadequately described.

10) The discussion of the inconsistencies between the three data types may be moved to the Discussion section.

Reviewer #3 (Remarks to the Author):

This study presents an ex vivo-digital microfluidic isolation of single cells for -Omics (evDISCO) platform for the analysis of astrocytes in response to stroke. This method is based on a platform named digital microfluidic Isolation of Single Cells for -Omics (DISCO; Nat. Commun., 2020, 11,5632). Key developments include updated interface and methods to select single cells from cryosectioned tissue slices, and allowed total automation of the sample preparation process. The method presented alongside two high-throughput platforms for spatial (Visium) and single-cell transcriptomics (10x Chromium) which may miss cell-type-specific nuances or spatial context. The authors concluded that, compared to these approaches, evDISCO allowed (1) specific selection of astrocytes at desired locations at single-cell resolution and (2) fully integrated device to gain reliability, benchmarked by distinguishing between proximal and distal astrocytes using both scRNA-seq and scProteomics analyses. Overall, this is an interesting manuscript introducing a promising approach and demonstrates a notable step forward for the field. Yet, there are few concerns and remarks in properly validating the platform, and generated data. A comment or comparison with other methods could also be useful to the readers.

Major comments:

- The main objective of this manuscript seems a bit obscure. Assuming the manuscript focuses on reporting the evDISCO platform, it will benefit from including more data and discussion regarding evDISCO's key features, characterizations, improvements, etc. Currently, its detailed discussion is not presented until Page 7 (& Fig. 4 and 5) in the manuscript. Indeed, most (if not all) content in the Figure 1-3 were data and analyses based on the Visium and 10x Chromium platforms. Meanwhile, if it focuses on new biological insights, there are also certain clarifications need to be provided (see following).

- Certain comparisons presented in the manuscript seems unclear and difficult to justified. For instance, why do authors use different markers to isolate astrocytes for subsequent analyses, and then compare? Specifically, ACSA-2 was used to isolate astrocytes when using the 10X Chromium method and analysis (Page 5, Line 132) while GFAP was chosen when using evDISCO (Page 7, Line 203). Is it possible that these may represent different astrocyte populations and thus cannot be directly compared?

- [Line 211-214] What is the basis of the classification of zones as proximal or distal (e.g., by location or by gene expression)? Is there a standard distance to determine these zones? The authors mentioned that the transcriptional differences between zones B and C surprised them. However, it is not clear how the authors classified both zones as proximal.

- [Line 301-302] Can the authors briefly explain how the difference in *Clu* expression via 10X Chromium and evDISCO are driven by a subset of proximal cells? Is there a particular reason why these cells would have detectable *Clu* via 10X Chromium and evDISCO but not Visium given its high spatial resolution?

- The proteomic digestion is performed for 19 h, which shall lead to substantial evaporation and thus possess adversely effects on final performance (especially on an open digital microfluidics).

Can the authors comment on this and how they manage to control such issue.

- A proper level of discussion can be helpful, especially towards the end of manuscript. Considering this is a promising platform, authors may elaborate more about its key advances, limitation and possible developments, such as how to tackle the overall lower throughput, and etc.

- The section "Integrating spatial and single cell transcriptomics identifies the heterogeneity of the astrocyte response in subacute stroke" has unclarities that may be improved.

a. [Line 148-149], the authors state that "Gene expression signatures in clusters 1-8 showed substantial overlap between the top 10 genes enriched in each cluster (Figure 3D)." However, Cluster 3 and Cluster 6 are visually different from other clusters.

b. Is the d10 sample (as shown in Fig. 3E) the same one used in Figs. 2A, 2C, 2E, and 5A? If so, please explain why the images (of Fig. 3E) are mirrored horizontally.

c. At Line 161 – 163, the Refs. 18 – 21 do not discuss the listed genes (Apoe, Flt1, Sparc).

Minor comments:

- Please explain how expression values are normalized.

- There are some typos in the text, and a repeating sentence at Line 133-136 (Page 5).

- Many parentheses appear in Methods, and some seem for unclear purpose.

- Line 199 mentioned a redesign to increase automation, however it is not clear in the manuscript how the platform is operated.

- The authors may provide control proteomic data done using a blank sample (negative control).

- Please specify the stroke lesion sites on Fig. 2A.

- Please annotate what factors (e.g., macrophage-dominant signature, glial signature) are shown in Fig. 2A.

- Does the first row of Fig.2E composed of all clusters? It is not labeled.

- [Line 91-95] Authors mention the experiment results; however, two rows of the data/images cannot be found on SI Fig. 2.

- Figure 5D and 5H, is the merged magenta color represents GFAP?

Responses to the reviewers' comments for manuscript NCOMMS-23-11404A

We thank all reviewers for their comments and have made changes to the manuscript accordingly, which are shown in blue lettering.

Reviewer #1 (Remarks to the Author):

Here the authors use a combination of single cell RNAseq (scRNAseq), spatial transcriptomics, and a recently published laser capture method the authors label 'DISCO' to investigate the gene and protein changes in cells following stroke. Of particular note, the 'evDISCO' proteomics seems to be single cell and is a highlight of the approaches used in this study.

The authors use these tools to define Apoe and Fabp7 as differentially regulated following cortical stroke.

Overall the manuscript uses modern methods, but these are underpowered such that it is unclear if the findings are appropriate. I have a number of other concerns with the manuscript that would benefit from clarification:

We thank the reviewer for these comments. We have taken your statement of underpowered studies very seriously and added two additional replicates per time point for Visium (total n = 3) and we have more than doubled the number of cells for 10X Chromium (now a combined total across two independent experiments of 19,055 cells).

1. Incorrect labeling of cell types (e.g. Figure 2c) - Vimentin is a gene that is highly expressed in astrocytes and endothelial cells, while several other genes are a greater vascular gene expression signal. How was this chosen as a purely macrophage signal?

Thank you for this comment. We have modified the text to explicitly state that macrophage markers (Lyz2 and Lgals3) were seen within the infarct spots, while astrocyte markers were seen in the border spots. Further, we have reviewed the text carefully to make sure that it does not state that all genes in the top panel of Fig 2C are macrophage markers.

This concern also follows to cell type signatures in 'Factors' in this figure - the 'glial factor signature' is a combination of multiple cell types - presumably astrocytes (Gfap, Apoe, etc.) microglia (Aif1, Trem2, C1a, etc.), and oligodendrocytes (e.g. Plp1). That this is so poorly defining a single cell type is problematic to the overall mechanism under investigation here.

We understand the Reviewer's concern about our cell type classification. As Visium cannot resolve data to a single cell level, we are cautious about overclassifying cell types. To address this concern, we have revised the text to explicitly state these are broad and generic signatures in the text. We further address the question of cell-type as we get into the more granular analyses that are possible with 10X Chromium and evDISCO.

2. Overinterpretation of results. The authors make bold statements that are not currently backed up by the data that is included. An example is at line 97-101 (end of the first

section) - these conclusions cannot be drawn from the data presented in this section of the manuscript. It is a hypothesis that needs to be tested, but there is no characterization of these statements.

We completely understand the concern here about cell type characterization. We have softened these statements in the text and highlighted that they are related to broad signatures captured by Visium (which **infers** what a cell/cell type may be expressing). Later, astrocytes are captured with evDISCO, which **directly** measures a cell and its expression with spatial context.

another example is in statements pertaining to the sequencing depth of 'evDISCO' - please provide details for 'higher sequencing depth' afforded using this method - if it is less than 50x that achieved by 10x chromium, the sequencing depth is actually shallower - as 50 cells have been combined for 'evDISCO' - which would place this method as a bulkRNAseq approach - which is most often deeper by virtue of a greater amount of genetic material being loaded into library preps and the sequencer. Note: the pooling of 50 cells for the 'evDISCO' transcriptomics methods seems to lower the threshold and sensitivity for these studies when compared to snRNAseq.

We appreciate the comment, which suggests that the original text was unclear. In fact, we do not combine 50 cells in the evDISCO data – evDISCO is inherently a single-cell technique, and all of the evDISCO data that we show is for single cells (not pooled). We have revised the text to make this clearer.

We note, for clarification, that “depth” refers to read counts per cell here. This increase in depth of reads per cell can be found in the metrics for 10X cells (Supplementary Figure 10), where the mean gene and read counts per cell are a little over 2000 and 5000, respectively. This is compared to evDISCO (Supplementary Figure 11), where mean gene and read counts per cell are a little over 5000 and 25,000, respectively.

3. astrocyte purification/enrichment (using ACSA2 antibody) - does ACSA2 enable the collection of all astrocyte subtypes, or is it only a sample of cortical astrocytes? How does the possible sub-sampling affect the conclusions drawn from this dataset as a result? Can the authors provide evidence that GLAST and ACSA2 label the same cells in the cortex? In the cerebellum these are highlighted as labeling overlapping by different populations (see PMID: 34060113). Atp1b2, the gene encoding ACSA2 has also been reported in neuronal cells, and in non-glial and glial progenitors.

This is an excellent point, that motivated us to perform a new set of IHC experiments. In contrast to previous work in the cerebellum, we found colocalization of ACSA2 and GLAST in the sensory motor cortex, which we have included in a new Supplementary Figure 4A.

We agree with the Reviewer’s point about diverse astrocyte populations, as we could potentially have selected for a [specific] population of ACSA2hi cells in our study, and as such, limited the types of astrocytes in our 10X Chromium data. Therefore, for the

Revisions experiments, we widened our ACSA2 FACS gates (see FACS plots in the new data in Supplementary Figure 4B) to capture the entire ACSA2+ fraction. While this led to the [surprise] capture of microglia, it also showed that **Atp1bp2** was co-expressed with other astrocytic genes, including *Aldh1l1*, **Slc1a3**, *Slc1a2*, and *Aldoc*, across astrocyte clusters 5,3,9,7, and 10 (**Figure 1**). This turned out to be a valuable addition to the paper and will also be an important technical note for the astrocyte community, and so, we thank the Reviewer for bringing this to our attention.

Figure 1. Violin plots showing the distribution of microglial (*Tmem119*, *P2ry12*, *Trem2*, & *Aif1*) and astrocytic (*Slc1a3*, *Slc1a2*, *Gfap*, *Aldoc*, *Aldh1l1*) genes (rows) across similarity-sorted clusters (columns, sorted by average gene expression) that are derived from the UMAP in Main Figure 3A. Expression (y-axis) values represent SCT-normalized gene expressions.

4. the only two transcripts/proteins identified as novel in this study at *ApoE* and *Fabp5* which have been investigated with some vigor in the stroke field. *ApoE* as an example was investigated by PMID: 19623195) and highlights a peri-infarct region astrocyte population that is *APOE* negative - the authors should discuss their results in the context of these, and other, historical data.

We agree with the reviewer. We have revised the manuscript to include a discussion of our results in the context of historical data in stroke and other neurodegenerative diseases, and more specifically, the peri-infarct astrocyte population mentioned by this Reviewer. This was an excellent addition and we thank the Reviewer for bringing this to our attention.

5. as a general statement, the new acronym 'DISCO' is poorly thought out, as this will cause confusion with iDISCO (immunolabeling-enabled three-dimensional imaging of solvent-cleared organs from the Tessier-Levigne group. In addition, the prefix of 'ex vivo' is incorrect - ex vivo refers to removal of a tissue/organ from the body for treatment and

then returning it to the body. In the context of this study, ex vivo is the incorrect term

We appreciate the point. Unfortunately, we need to stick with 'DISCO' for consistency because the technique was introduced with this name in our 2020 paper¹ (which has now been cited >70 times). The potential for confusion with the Tessier-Levigne method is a real one, and with this in mind we have added a new sentence to the introduction citing the Tessier-Levigne paper and clarifying the difference between the two techniques.

As for the prefix – we need a name that distinguishes the new technique (which evaluates tissue biopsy slices that are loaded, intact, onto the device) from the original DISCO technique (which evaluated cells that were loaded as suspension, adhered, and then cultured *in vitro*).

We agree that “ex vivo” is not perfect, and in the revised manuscript, we have renamed the new technique to be “tissue-DISCO” or “t-DISCO” in short. We thank the reviewer for helping us tune the name to be more precise.

6. can the authors comment on the alarming loss of Apoe in subcortical regions >2 days following stroke? This seems to be the most striking change in expression that is maintained for at least 3 weeks.

We hypothesize that this is a by-product of Visium technology. Similar to the other candidate genes, *Clu* and *Aldoc*, *Apoe* may just not be among the highest expressed genes in the infarct area at day 2 post-stroke (we do not believe, yet, that *Apoe* disappears at day 2). We have modified the text to include a discussion of the differences in gene detection methods of [now termed] tDISCO relative to Visium and 10X Chromium technologies to address this comment.

7. underpowered studies - this is a major concern, as the validation steps are minimal (2 genes) and therefore the scRNAseq and spatial data for the basis of most of this manuscript.

scRNAseq: the cell numbers collected for scRNAseq studies are incredibly low: 3009 (uninjured, from 8 brains, ~376 cells/brain) and 2529 (stroke, from 8 brains, ~316 cells/brain).

spatial transcriptomics: only a single section is completed for each group of animals - giving an n of 1.

These both seems very underpowered and are thus unlikely to be sufficiently able to address the specific questions the authors are attempting to.

As mentioned above, to address this important concern, we sequenced an additional 13,517 ACSA-2+ cells from injured (6243 cells) and non-injured (7274 cells) for a total of 19,212 cells analyzed across two independent experiments. We also added two more replicates to each Visium timepoint, for an n=3 (biological replicates) per timepoint.

We also validated a total of five genes with RNAScope. Two genes are now shown in the main figure, *Apoe* and *Clu*. Three additional genes are shown in the Supplementary

Information, *Cd81*, *Fabp5*, and *Aldoc*. Also to note, *Apoe*, *Fabp5*, and *Cd81* were found across platforms, and in this way each platform serves to 'validate' the other.

Minor points:

1. Figure 2, panel C - (in relation to color scale) What is 'value' referring to? Raw expression values? Normalized values?

Our apologies for this oversight. "The value represents the summed SCT normalized expression values corresponding genes from (B)." This has been added to the figure caption.

2. Figures (in general) - the manuscript and figures are difficult to follow (e.g. line 222 'Figure 4G, third group down'). The authors might consider a way to more clearly define these figures - perhaps labeling the individual 'groups' running down the second panel included in G and H (for example). This is a concern for many of the figures.

We have provided additional labels to clarify these groups in Figure 4

Panel A in Figure 4 offers considerably little detail about the 'evDISCO' method - as a reviewer I had to read the recent manuscript from Lamanna in Nat Comm to understand what exactly had been done.

We agree that we relied heavily on our previous publication to explain the [now termed] tDISCO method. To address this concern, we have now added the following description to the manuscript:

"The tDISCO system makes use of a laser to create cavitation bubbles that mechanically release selected single cells into the receiving buffer above the tissue, while digital microfluidics is used to collect this single cell and replenish the buffer droplet above the tissue, allowing for iterative lysis of individual cells from a single tissue." To the tDISCO results section.

Additionally, the features novel to this newer version of DISCO include: a new microscope and digital microfluidic-capable stage (pictured in Figure 4A, with the main differences labeled in the diagram), a new laser source and integration strategy, a new LED fluorescent source, and a newer camera for imaging."

3. Figure 5, panels E-H - what time following stroke are these images taken?

Our apologies for this oversight. These were taken at d10 post-stroke and are now indicated in the figure legend

Reviewer #2 (Remarks to the Author):

The authors developed a platform called evDISCO for spatially resolved single-cell

transcriptomics and proteomics. In order to examine astrocyte diversity in stroke, they utilized evDISCO alongside two high-throughput platforms for spatial (Visium) and single-cell transcriptomics (10x Chromium). Integrating Visium and 10X Chromium datasets, they identified two astrocyte populations, either proximal or distal to the injury site. Then, evDISCO enabled further refinement of the spatial boundaries that define these populations and revealed their molecular profiles. This study provided a significant resource for elucidating the roles of astrocytes in stroke. However, the article requires some revisions.

We thank Reviewer 2 for the comments and appreciate the attention to detail. We have fixed all the errors mentioned and hope we have addressed all concerns below.

1) It would be helping to provide a more comprehensive explanation for the selection of the time points 2, 10, and 21 days.

Thank you for this comment. Human stroke is characterized by four phases hyperacute, acute, subacute and chronic, which represent different pathophysiological processes. Hyperacute stroke is characterized by the disruption of the blood brain barrier and clot formation (not represented in rodent models); acute stroke is a period of neuroinflammation characterized by ongoing damage. In the subacute and chronic phases there is neurorepair, including synaptic plasticity, neoangiogenesis and neurogenesis². Rodent models of ischemic events can mimic these pathologies³ and the timelines of these pathological events in preclinical models roughly occur within the following timelines⁴⁻⁶: acute stroke (24h-4d), subacute stroke (4d-21d), and chronic stroke (21d+). Therefore, we selected 2d, 10d and 21d to represent the different stroke stages in this study and have added this rationale to the manuscript.

2) The article has a number of minor errors. For instance, the illustration in Supplemental Figure 2 shows there are three rows, but only the third row is visible. And reference 4 is a 2020-accepted article; therefore, a volume and issue number must be provided.

Our apologies, these have been fixed.

3) For the conclusion of Figure 3B, namely that the proportion of each cell subtype differs between stroke and control, is there any way to determine if this is due to randomness, given that there appears to be only one sample for stroke and control?

This was an excellent question. The original stroke and uninjured samples represented pools of 8 mice per treatment condition. Biological pools incorporate individual mouse variability, and therefore, a signal arising from this data should demonstrate some synchrony across the 8 individual mice.

To further address these concerns, we sequenced an additional stroke and control group, each comprised of a biological pool of 12 mice. In total this added 15,000 more cells (13,000 after filtering) to our manuscript. Important, the new control and stroke datasets showed similar trends, which reflects the robust nature of these candidate genes,

above randomness. In addition, the use of complementary datasets from distinct platforms and additional validation of the 5 candidate gene targets with RNAScope further demonstrates the robustness of our main findings.

4) How can the influence of chance on the conclusions be excluded when there appears to be a single sample for all three types of data? As the authors note later, there are also inconsistencies between the three types of data; are these inconsistencies the result of randomness?

As described above (#3), and in response to Reviewer 1 (Comment 7), we added two more biological replicates to each Visium analysis (for a total of n=3 per time point) and more than doubled the number of cells analyzed for 10X Chromium.

5) How were the genes *Fabp5*, *ApoE*, and *Cd81* identified from the 1,221 differentially expressed genes?

To clarify, we first chose to focus on candidates from our 10X Chromium analyses, we then overlaid these genes back onto our d10 Visium section to check for congruence. Although we did find some genes such as *ApoE*, *Fabp5*, and *Cd81* that showed varying levels of congruence across all three platforms, we also found genes such as *Clu* and *Aldoc* that had conflicting representation across the three platforms. We have included a more thorough explanation in the “*Candidate gene inconsistencies across three sequencing platforms*” section of the Results.

6) The pixels in Supplemental Figure 4 are too small and do not appear to effectively convey the most important platform characteristics and optimization information.

Our apologies for the low resolution of the original figure, we have now corrected this in the revised submission.

7) The article fails to introduce and explain the distinctions between zones A, B, C&D, despite the fact that zone A is closer to the injury site than zone B. Could this be discussed or explained? In addition, is there any functional overlap or similar enrichment between the differential expression of proximal and distal astrocytes identified by evDISCO and the differential expression identified by 10x Chromium scRNA-seq data?

We appreciate this point. We have modified the text to explain that the zones were selected purely based on the relative distance to the stroke injury site. This was an unbiased approach that allowed us to bin cells into Zone A, closest to the injection site (needle track), and then at increasing standard 200uM intervals. In Supplementary Figure 8, we analyzed the gene expression differences in zones A relative to B, C, and D (versus B relative to A, C and D, which are shown in Main Figure 4). These results are reported in the Results section “*Spatially resolved single-cell transcriptomics with tDISCO defines proximal and distal astrocytes.*”

We also rewrote the Results section (*Candidate gene inconsistencies across three sequencing platforms*) to specifically address the overlap of findings between the Visium, 10X, and evDISCO platforms.

8) According to the 10x Chromium scRNA-seq data, the genes *ApoE* and *Fabp5* are enriched in astrocytes proximal to the lesion site. The authors present the expression of mRNA in the evDISCO data; are expression data for proteins available?

Unfortunately, these genes were not detected in the proteomics data. We have confirmed FABP5 expression in proximal cells and are currently investigating APOE as part of an ongoing follow-up study of proximal astrocytes.

9) The description of Main Figure 5 in "Results" is insufficient. The results in the "ApoE and Fabp5 are enriched in astrocytes proximal to the lesion site" are inadequately described.

Thank you, we agree. We have re-written the Results section "*Candidate gene inconsistencies across three sequencing platforms*" that corresponds to Main Figure 5 to better represent the data. Specifically, we have now described a detailed comparison of candidate genes across all platforms, with RNAScope serving as an ultimate validation/"tie breaker".

10) The discussion of the inconsistencies between the three data types may be moved to the Discussion section.

Thank you for this comment. We agree and have moved the bulk of the discussion of the differences between the technologies to the Discussion.

Reviewer #3 (Remarks to the Author):

This study presents an ex vivo-digital microfluidic isolation of single cells for -Omics (evDISCO) platform for the analysis of astrocytes in response to stroke. This method is based on a platform named digital microfluidic Isolation of Single Cells for -Omics (DISCO; Nat. Commun., 2020, 11,5632). Key developments include an updated interface and methods to select single cells from cryosectioned tissue slices and allowed total automation of the sample preparation process. The method presented alongside two high-throughput platforms for spatial (Visium) and single-cell transcriptomics (10x Chromium) which may miss cell-type-specific nuances or spatial context. The authors concluded that, compared to these approaches, evDISCO allowed (1) specific selection of astrocytes at desired locations at single-cell resolution and (2) fully integrated device to gain reliability, benchmarked by distinguishing between proximal and distal astrocytes using both scRNA-seq and scProteomics analyses. Overall, this is an interesting manuscript introducing a promising approach and demonstrates a notable step forward for the field. Yet, there are few concerns and remarks in properly validating the platform, and generated data. A comment or comparison with other methods could also be useful to the readers.

We thank Reviewer 3 for their comments and their interest in the platform. We appreciate their perspective that the objective of the paper was not clear and have re-worked the manuscript to better compare the platforms and focus the overall objective on technology development (for evDISCO). We included more comprehensive comparisons in the discussion to highlight how the technologies were used in a complementary fashion, but how this also serves as a measure for comparison.

Major comments:

- The main objective of this manuscript seems a bit obscure. Assuming the manuscript focuses on reporting the evDISCO platform, it will benefit from including more data and discussion regarding evDISCO's key features, characterizations, improvements, etc. Currently, its detailed discussion is not presented until Page 7 (& Fig. 4 and 5) in the manuscript. Indeed, most (if not all) content in the Figure 1-3 were data and analyses based on the Visium and 10x Chromium platforms. Meanwhile, if it focuses on new biological insights, there are also certain clarifications that need to be provided (see following).

We thank Reviewer 3 for this comment. We agree that the original text did not make clear the relationship (and advantages/disadvantages) of evDISCO relative to Visium and 10X Chromium technologies. We have re-written the results section associated with Main Figure 5, as well as the discussion.

- Certain comparisons presented in the manuscript seem unclear and difficult to justify. For instance, why do authors use different markers to isolate astrocytes for subsequent analyses, and then compare? Specifically, ACSA-2 was used to isolate astrocytes when using the 10X Chromium method and analysis (Page 5, Line 132) while GFAP was chosen when using evDISCO (Page 7, Line 203). Is it possible that these may represent different astrocyte populations and thus cannot be directly compared?

We appreciate the point and have attempted to improve the clarity of the comparisons throughout the manuscript. Regarding the specific question: We chose ACSA-2 to isolate astrocytes for 10X Chromium, as ACSA-2 was previously reported as a 'first-choice' astrocyte cell surface marker for downstream applications across models and injury paradigms⁷. Likewise, ASCA-2 was used in previous single-cell RNAseq studies^{8,9}. In sum, our choice of ACSA-2 selection for 10X Chromium was to ensure capture of a broad population of astrocytes within the stroke-injured cortex.

We agree with the reviewer that it would have been ideal to also use ASCA-2 to isolate cells with evDISCO. Unfortunately, ACSA-2 staining is diffuse and single cell bodies are impossible to distinguish by microscopy (see Figure 2 below). Because GFAP is known to be upregulated in stroke¹⁰ and we confirmed that *Gfap* expression was present in the injured cortex in our Visium dataset (Figure 3, below), we used GFAP to identify individual astrocytes for downstream evDISCO-based -omics. With this in mind, we carried out new experiments to validate GFAP expression in ACSA2+ astrocytes (Figure 2, below; white

boxes show examples of GFAP+ACSA2+ astrocytes). In addition, our 10X data shows expression of GFAP in all of our clusters (except cluster 6 denoted as VLMCs). Moreover, as shown in Main Figure 3E&G, GFAP is particularly enriched in Cluster 3, representative of our 'proximal' astrocytes, and therefore also supports comparison of astrocyte data across these platforms, despite differences in cell marker selection.

Finally, as described in response to Reviewer 1 (Comment 3) we also took into consideration this point about diverse astrocyte populations (from both Reviewers 1 and 3)

Figure 2. IHC of GFAP (green) and ACSA-2 (red) in the d10 stroke injured cortex. **A)** schematic of imaged regions from proximal to distal regions within the injured cortex. **B)** GFAP+ astrocytes co-express ACSA2. White boxes show examples of GFAP+ACSA2+ cells. Rows represent increasing proximal to distal distances, and match box colors in schematic.

Figure 3. *Gfap* expression in the d10 Visium dataset. Colour scale and value represents SCT-transformed *Gfap* expression value overlaid onto d10 Visium section.

- [Line 211-214] What is the basis of the classification of zones as proximal or distal (e.g., by location or by gene expression)? Is there a standard distance to determine these zones? The authors mentioned that the transcriptional differences between zones B and C surprised them. However, it is not clear how the authors classified both zones as proximal.

This is an excellent question. The zones were defined by their distance from the infarct core (as measured by the visible needle injection). Using an unbiased approach, a standard interval distance of 200uM was used to bin cells. This allowed us to accommodate for variation between sections (where the needle tract could be in slightly different places), and for biological variation (between different mice). The classification of 'proximal' versus 'distal' was based on gene expression and also guided by the expression patterns seen in the 10X Chromium analysis.

- [Line 301-302] Can the authors briefly explain how the difference in *Clu* expression via 10X Chromium and evDISCO are driven by a subset of proximal cells? Is there a particular reason why these cells would have detectable *Clu* via 10X Chromium and evDISCO but not Visium given its high spatial resolution?

This is an excellent question that we will build off of in future studies. We hypothesize that *Clu* is driven by a subset of proximal cells because evDISCO data shows that only half of 'proximal' cells express *Clu*. In contrast, RNAScope shows no obvious difference in *Clu* expression in GFAP+ proximal and distal cells. Therefore, these differences could result from gene expression in a subset of cells that were captured with evDISCO but are not obvious when examining the entire astrocyte population with RNAScope. We have also clarified in the paper that this could be a potential pitfall of evDISCO, whereby cell subsampling could lead to misrepresentation of the occurrence of the GFAP+*Clu*+ astrocytes in the proximal space.

- The proteomic digestion is performed for 19 h, which shall lead to substantial evaporation and thus possess adverse effects on final performance (especially on open digital microfluidics). Can the authors comment on this and how they manage to control

such issue.

We agree with the Reviewer and understand the confusion. In some papers (e.g., *Chem. Sci.*, 2023, 14, 2887-2900), we and others have described fully integrated methods in which the proteomic sample handling is carried out in the device itself, which indeed requires careful attention to digestion. But in this paper, we elected to collect the lysates and run the processing steps (including the proteomic digestion) off-chip, in capped tubes. We have revised the text to make this clearer.

- A proper level of discussion can be helpful, especially towards the end of manuscript. Considering this is a promising platform, authors may elaborate more about its key advances, limitation and possible developments, such as how to tackle the overall lower throughput, and etc.

We completely agree and have made improved the discussion to better elaborate key advances with evDISCO and how future efforts will be geared towards addressing throughput, the manual nature of cell selection, and better integration of layers of data onto a single cell.

- The section “Integrating spatial and single cell transcriptomics identifies the heterogeneity of the astrocyte response in subacute stroke” has unclarities that may be improved.

- a. [Line 148-149], the authors state that “Gene expression signatures in clusters 1-8 showed substantial overlap between the top 10 genes enriched in each cluster (Figure 3D).” However, Cluster 3 and Cluster 6 are visually different from other clusters.
- b. Is the d10 sample (as shown in Fig. 3E) the same one used in Figs. 2A, 2C, 2E, and 5A? If so, please explain why the images (of Fig. 3E) are mirrored horizontally.

In the new manuscript, we have made the UMAP plots consistent between Main Figures 3 and 5. Please note that upon re-analysis (of the new data) these clusters changed slightly. The Visium sections are the same d10 sections between Figures 2, 3, and 5. In Figure 2 we have just utilized the WarpImages() function of STUtility package to get the mirror image and have all the sections in the same orientation.

- c. At Line 161 – 163, the Refs. 18 – 21 do not discuss the listed genes (ApoE, Flt1, Sparc).

We appreciate the keen eye of the reviewer, which helped us correct this embarrassing mistake – there was an error in the final bibliographic compilation in the original manuscript. In the new submission, we have carefully reviewed each reference in the manuscript and confirmed they are matched to the correct information, including the above example.

Minor comments:

- Please explain how expression values are normalized.

We apologize for the oversight; this has now been specified in each of the figure captions. All expression values were log-normalized using the package Seurat. We then used SCTransform(), which calculates technical noise in the dataset (using negative binomial regression) and uses the residuals of this model as the normalized values. A more in-depth description can be found here:

https://satijalab.org/seurat/articles/sctransform_vignette.html

- There are some typos in the text, and a repeating sentence at Line 133-136 (Page 5).

We apologize, these have been corrected.

- Many parentheses appear in Methods, and some seem for unclear purpose.

We apologize for this –we were referring functions within a software package that include parentheses in the naming convention. We agree that this makes the text unwieldy and thus have removed these parentheses in the revised text.

- Line 199 mentioned a redesign to increase automation, however it is not clear in the manuscript how the platform is operated.

We understand this criticism and have added a sentence to clarify how the system works in Line 199.

- The authors may provide control proteomic data done using a blank sample (negative control).

We appreciate the suggestion and have added a new dataset (in Supplementary Figure 12C) that better describes the sensitivity of our proteomics pipeline.

- Please specify the stroke lesion sites on Fig. 2A.

Here, we are relying on the clustering in Figure 2E to best demarcate the stroke injury site as defined by molecular markers.

- Please annotate what factors (e.g., macrophage-dominant signature, glial signature) are shown in Fig. 2A.

- Does the first row of Fig.2E composed of all clusters? It is not labeled.

Yes it does, we have made sure to include that in the figure caption.

- [Line 91-95] Authors mention the experiment results; however, two rows of the data/images cannot be found on SI Fig. 2.

We are sorry about this. This [upload] issue has been corrected.

- Figure 5D and 5H, is the merged magenta color represents GFAP?

The magenta colour represents colocalization of GFAP and the candidate gene of interest. This has now been clarified in the figure caption.

References

1. Lamanna J, Scott EY, Edwards HS, et al. Digital microfluidic isolation of single cells for -Omics. *Nature Communications* 2020 11:1. 2020;11(1):1-13. doi:10.1038/s41467-020-19394-5
2. Bernardo-Castro S, Albino I, Barrera-Sandoval ÁM, et al. Therapeutic Nanoparticles for the Different Phases of Ischemic Stroke. *Life* 2021, Vol 11, Page 482. 2021;11(6):482. doi:10.3390/LIFE11060482
3. Carmichael ST. Rodent models of focal stroke: size, mechanism, and purpose. *NeuroRx*. 2005;2(3):396-409. doi:10.1602/NEURORX.2.3.396
4. Roome RB, Bartlett RF, Jeffers M, Xiong J, Corbett D, Vanderluit JL. A reproducible Endothelin-1 model of forelimb motor cortex stroke in the mouse. *J Neurosci Methods*. 2014;233:34-44. doi:10.1016/J.JNEUMETH.2014.05.014
5. Dai PM, Huang H, Zhang L, et al. A pilot study on transient ischemic stroke induced with endothelin-1 in the rhesus monkeys. *Scientific Reports* 2017 7:1. 2017;7(1):1-12. doi:10.1038/srep45097
6. Abeysinghe HCS, Bokhari L, Dusting GJ, Roulston CL. Brain Remodelling following Endothelin-1 Induced Stroke in Conscious Rats. *PLoS One*. 2014;9(5):e97007. doi:10.1371/JOURNAL.PONE.0097007
7. Batiuk MY, De Vin F, Duqué SI, et al. An immunoaffinity-based method for isolating ultrapure adult astrocytes based on ATP1B2 targeting by the ACSA-2 antibody. *Journal of Biological Chemistry*. 2017;292(21):8874-8891. doi:10.1074/jbc.M116.765313
8. Batiuk MY, Martirosyan A, Wahis J, et al. Identification of region-specific astrocyte subtypes at single cell resolution. *Nature Communications* 2020 11:1. 2020;11(1):1-15. doi:10.1038/s41467-019-14198-8
9. Lattke M, Goldstone R, Ellis JK, et al. Extensive transcriptional and chromatin changes underlie astrocyte maturation in vivo and in culture. *Nat Commun*. 2021;12(1). doi:10.1038/S41467-021-24624-5
10. Hol EM, Pekny M. Glial fibrillary acidic protein (GFAP) and the astrocyte intermediate filament system in diseases of the central nervous system. *Curr Opin Cell Biol*. 2015;32:121-130. doi:10.1016/J.CEB.2015.02.004

REVIEWERS' COMMENTS

Reviewer #1 (Remarks to the Author):

The authors has addressed my earlier concerns, and I thoroughly enjoyed reading both the updated manuscript, and the thoughtful responses to both the points raised by this reviewer, as well as those raised by the other two reviewers.

The additional visium experiments to increase n values/power, careful descriptions of additional details, and the expanded discussion section are great additions.

No additional comments from me.

Reviewer #2 (Remarks to the Author):

The authors have addressed all of my concerns.

Reviewer #3 (Remarks to the Author):

I appreciate the authors for making a great effort in addressing concerns and comments by adding data/new analysis and explanatory text. Especially the revised section titled "Candidate gene inconsistencies across three sequencing platforms" provides a clearer discussion concerning pros and cons among different platforms is very helpful. Following are remaining small remarks:

> To avoid confusion among readers, it is crucial to clarify that the assertion claiming tDISCO functions as a platform capable of carrying out spatial sc-OMICS, especially regarding its sensitivity for single-cell proteomics profiling. For instance, the off-chip proteomics operation had led to poor sensitivity in protein identification coverage (Supplementary fig. 13). This discrepancy is apparent when compared such results (e.g., 0.25 ng) to other similar spatial single-cell proteomics platforms.

> Minor point: there are still certain text written as "evDISCO" that need change.

Responses to the reviewers' comments for manuscript NCOMMS-23-11404A

We thank all the Reviewers for their final comments. We have made final changes to the manuscript to address the remaining minor concerns, which are shown in **red lettering**.

Reviewer #1 (Remarks to the Author):

The authors has addressed my earlier concerns, and I thoroughly enjoyed reading both the updated manuscript, and the thoughtful responses to both the points raised by this reviewer, as well as those raised by the other two reviewers.

The additional visium experiments to increase n values/power, careful descriptions of additional details, and the expanded discussion section are great additions.

No additional comments from me.

Reviewer #2 (Remarks to the Author):

The authors have addressed all of my concerns.

Reviewer #3 (Remarks to the Author):

I appreciate the authors for making a great effort in addressing concerns and comments by adding data/new analysis and explanatory text. Especially the revised section titled "Candidate gene inconsistencies across three sequencing platforms" provides a clearer discussion concerning pros and cons among different platforms is very helpful. Following are remaining small remarks:

> To avoid confusion among readers, it is crucial to clarify that the assertion claiming tDISCO functions as a platform capable of carrying out spatial sc-OMICS, especially regarding its sensitivity for single-cell proteomics profiling. For instance, the off-chip proteomics operation had led to poor sensitivity in protein identification coverage (Supplementary fig. 13). This discrepancy is apparent when compared such results (e.g., 0.25 ng) to other similar spatial single-cell proteomics platforms.

We thank the reviewer for the comment, and agree that the total numbers of proteins per cell reported here is lower than the state-of-the-art. We propose that this is because of the mass spectrometer used herein, an Orbitrap QE HF, which was acquired in 2018. The reviewer likely knows that mass spectrometer sensitivities have soared in recent years, such that recent papers using the latest generation of mass spectrometers are reporting more than 1,000 proteins per cell (Brunner et al. Mol. Sys. Biol. 2022, 18, e10798). We expect that this limitation is a function of using an older mass spectrometer and look forward to testing this hypothesis in the future.

> Minor point: there are still certain text written as "evDISCO" that need change.

We thank the reviewer for the comment, our apologies for the oversight. This has been corrected.